# Toward Foundation Model for Multivariate Wearable Sensing of Physiological Signals

## Abstract

Time-series foundation models have the ability to run inference, mainly forecasting, on any type of time series data, thanks to the informative representations comprising waveform features. Wearable sensing data, on the other hand, contain more variability in both patterns and frequency bands of interest and generally emphasize more on the ability to infer healthcare-related outcomes. The main challenge of crafting a foundation model for wearable sensing physiological signals is to learn generalizable representations that support efficient adaptation across heterogeneous sensing configurations and applications. In this work, we propose NormWear, a step toward such a foundation model, aiming to extract generalized and informative wearable sensing representations. NormWear has been pretrained on a large set of physiological signals, including PPG, ECG, EEG, GSR, and IMU, from various public resources. For a holistic assessment, we perform downstream evaluation on 11 public wearable sensing datasets, spanning 18 applications in the areas of mental health, body state inference, biomarker estimations, and disease risk evaluations. We demonstrate that NormWear achieves a better performance improvement over competitive baselines in general time series foundation modeling. In addition, leveraging a novel representation-alignment-match-based method, we align physiological signals embeddings with text embeddings. This alignment enables our proposed foundation model to perform zero-shot inference, allowing it to generalize to previously unseen wearable signal-based health applications. Finally, we perform nonlinear dynamic analysis on the waveform features extracted by the model at each intermediate layer. This analysis quantifies the model's internal processes, offering clear insights into its behavior and fostering greater trust in its inferences among end users.

## 1 Introduction

Mobile and wearable sensors have been shown to be valuable for the field of healthcare by passively and continuously tracking physiological signals such as photoplethysmography (PPG) for pulse, electrocardiography (ECG) for heart activity, galvanic skin response (GSR), and electroencephalography (EEG) for brain activity. These time series signals are beneficial for early diagnosis, personalized health insights, and remote patient monitoring (Zhang et al., 2024a).

Recently, various foundation models on time series have been proposed (Ansari et al., 2024; Abbaspourazad et al., 2023; Woo et al., 2024; Foumani et al., 2024). Another common approach for signal modeling involves converting raw signal series into 2D images or spectrograms, using fixed-size sliding windows, followed by the use of visual encoders like Vision Transformers (ViT) to extract representations for making inferences (Semenoglou et al., 2023; Wimmer & Rekabsaz, 2023; Vishnupriya & Meenakshi, 2018; Chun et al., 2016; Krishnan et al., 2020; Dosovitskiy et al., 2020). These works have significantly advanced the field and provided valuable insights, yet two main issues still exists which need further exploration to fully understand their potential in wearable scenarios. First, contrastive learning-based foundation models (Abbaspourazad et al., 2023) rely on a predefined set of input signal types, making them unsuitable when transferring to scenarios with different types and numbers of sensors. Second, while both time series foundation models (Ansari et al., 2024; Zhang et al., 2022; Woo et al., 2024) and spectral-based approaches (Semenoglou et al., 2023; Wimmer & Rekabsaz, 2023) attempt to address this issue by training a generic encoder that can handle type-agnostic series, they remain limited to processing only univariate series. Because of

Figure 1: The role of our framework. Several icons from Freepik (n.d.); Zhang et al. (2024a).)

this constraint, these previous works fail to account for the heterogeneity of multivariate input data; specifically, they do not capture the complex relationships between signals from sensors located on different body parts. These two limitations of recent approaches hinder their generalization and usefulness for wearable health monitoring.

Moreover, Wearable-based multimodal physiological signals present unique challenges that distinguish them from general time series data, such as stock prices or weather patterns. Wearable signal modalities, such as PPG and EEG, vary in characteristics like dimensionality, sampling rate, and resolution, often requiring modality-specific preprocessing. Existing methods tokenize raw signals (Ansari et al., 2024; Zhang et al., 2022) or convert them into image or spectral representations (Wu et al., 2023; Mathew et al., 2024; Vaid et al., 2023). While effective for specific tasks, these approaches lack generalizability and fail to provide a consistent preprocessing pipeline across multiple modalities. A consistent framework that accommodates diverse signal requirements is essential for training deep learning-based foundation models and advancing multimodal signal analysis. Finally, digital healthcare applications emphasize model interpretability and robustness, which reveals an unignorable research gap in recent literature on studying the intrinsic behaviors of their proposed models.

In this work, we present NORMWEAR, a normative foundation model, aiming to learn effective wearable sensing representations, addressing the above-discussed research gaps. NORMWEAR has been pretrained on more than 2.5 million multivariate wearable sensing segments, comprising total of 14,943 hours of sensor signal series, using publicibly avaliable datasets. We evaluated NORMWEAR on 18 public downstream tasks against competitive baselines under both linear probing and zero-shot inference. Overall, our contributions with the proposed NORMWEAR healthcare modeling framework can be summarized as follows:

- To our knowledge, we are the first to develop a foundation model specifically designed for wearable sensing data, capable of processing arbitrary configuration of multivariate signals from sources such as the heart, skin, brain, and physical body.

- NORMWEAR comprises novel methodologies built upon the advanced practice in both the fields of signal processing and deep learning, including (a) continuous wavelet transform (CWT) based multi-scale representations for modality- and number-agnostic tokenization, (b) channel-aware attention layer that enables the model to process arbitrary multivariate inputs, and (c) zero-shot inference with human sensing adapted fusion mechanism for improved efficacy.

- We are also the first to integrate and process a comprehensive wearable signals dataset with varied number of input channels for training self-supervised learning algorithms, with thorough downstream evaluation. These datasets cover key health applications, including mental and physical state inference, biomarker estimation, and disease risk evaluation. We make the preprocessed data, codebase, and model weights publicly available.

- We perform a comprehensive interpretability analysis and visualization to elucidate the model's inner workings and decision-making processes, and we are the first to quantify the analysis with nonlinear-dynamic-analysis of the waveform features extracted by the models

at each intermediate layer, offering insights into NORMWEAR's neural activity patterns across various sensing signal types and tasks. This is crucial for validating the reliability of downstream applications and building trust with end users.

Our proposed NORMWEAR aims to provide a generalized data representation solution for smart health monitoring, benefiting the general public, and serving as a fundamental tool for researchers and professionals to address future healthcare challenges.

## 2 METHOD

Table 1: Downstream evaluation data. All these data are unseen during pretraining.

| Downstream Dataset | Sensor | Tasks | #Samp. (#Subj.) |
|---|---|---|---|
| WESAD (Schmidt et al., 2018) | IMU, PPG, ECG, GSR | Stress Detection | 11050(15) |
| UCI-HAR (Reyes-Ortiz et al., 2012) | IMU | HAR | 10299(30) |
| DriverFatigue (Min et al., 2017) | EEG | Fatigue Detection | 2400(12) |
| **Activity Recognition Total** | - | - | **23749(57)** |
| Epilepsy (Andrzejak et al., 2023) | EEG | State Recognize | 11500(500) |
| GAMEEMO (Alakus et al., 2020) | EEG | Valence-Arousal | 5600(28) |
| **EEG Main Tasks Total** | - | - | **17100(528)** |
| ECG-Abnormal (Bousseljot et al., 2009) | ECG | Abnormal Detection | 11640(249) |
| PPG-BP (Liang et al., 2018) | PPG | Risk of Diseases | 657(219) |
| PhysioNet EMG (Goldberger et al., 2000) | EMG | Muscular Diseases | 163(3) |
| **Risk Evaluation Total** | - | - | **12460(471)** |
| Noninvasive-BP (Esmaili et al., 2017) | PPG | BP Estimate | 125(26) |
| PPG-Hgb (Esmaili et al., 2017) | PPG | Hgb Estimate | 68(68) |
| Fetal-fPCG (Bhaskaran et al., 2022) | PCG | Fetal HR Estimate | 47(47) |
| **Vital Signs Total** | - | - | **240(141)** |
| **Total All** | - | - | **53549(1197)** |

Table 2: Baselines and pretraining data.

| Baseline Methods | Modeling Strategies | |
|---|---|---|
| TF-C (Zhang et al., 2022) | SoTA in TS SSL; modeling time and frequency domain information at same time. | |
| CLAP (Wu et al., 2023) | SoTA in audio modeling; process signal as spectrogram | |
| Chronos (Ansari et al., 2024) | SoTA in TS forecasting, leverage LLM for modeling | |
| Statistical approach | Reserve full interpretability | |

| Pretrain Dataset | Sensors | #Samp (hours). |
|---|---|---|
| Cuff-Less-BP (Kachuee et al., 2016) | ECG, PPG | 42934(72) |
| PPG-Dalia (Reiss Attila, 2019) | ECG, PPG IMU, GSR | 42606(71) |
| Auditory-EEG (Alzahab et al., 2022) | EEG | 13601(23) |
| PhyAAt (Bajaj et al., 2020) | EEG | 19550(33) |
| MAUS (Beh et al., 2021) | ECG, PPG GSR | 13068(22) |
| Mendeley-YAAD (Dar et al., 2022) | ECG, GSR | 2964(5) |
| Brain-Cognitive (Dar et al., 2022) | EEG | 51201(85) |
| EPHNOGRAM (Dar et al., 2022) | ECG, PCG | 36611(61) |
| BIDMC (Dar et al., 2022) | ECG, PPG | 8427(14) |
| **Num Segments (# Segm.)** | - | **230,962(385)** |
| **# Segm. w/ Augment** | - | **2,576,418(4,294)** |
| **Num Sensor Signals (# Sign.)** | - | **802,019(1,337)** |
| **# Sign. w/ Augment** | - | **8,965,538(14,943)** |

### 2.1 DATASET CONSTRUCTION FOR MODEL PRE-TRAINING AND DOWNSTREAM EVALUATION

We curated a collection of 9 publicly available datasets (Table 2) exclusively for model pre-training, resulting in approximately 230,962 multivariate time series segments, comprising 4,294 hours of total sensor signal series, across various modalities, including PPG, ECG, EEG, GSR, PCG, and inertial measurement unit (IMU) data. To address the dataset size limitation, we then applied herustic data augmentation (algorithm 1) to expand the pre-train dataset to 2.5 million segments, comprising 14,943 hours of total sensor signal series. Notably, each sample segment may contain a variable number of input channels depending on the sensor signals provided by the respective datasets. This input configuration aligns seamlessly with our model's design, which is optimized to flexibly handle arbitrary numbers and configurations of sensor signal inputs.

To prevent potential data leakage in downstream tasks, we evaluate our model's transferability using an additional 11 publicly available datasets encompassing 18 modeling tasks, which include affective state classification, physical state recognition, biological estimation, and disease risk evaluation. Details about the datasets is presented in Table 1.

### 2.2 TOKENIZATION

Tokenization is a fundamental term widely used in natural language processing. In the context of wearable sensing, we leverage this term to represent the stage of signal processing before sending the processed data to the deep learning-based encoder. Spectral methods, which utilize the short-time Fast Fourier Transform (FFT) (Brigham, 1988) with a sliding window to compute spectrograms, are widely regarded as the benchmark approach for tokenization. However, due to the inherent trade-off between time and frequency resolution, the spectral representation with a fixed window size cannot be generalized. This is because the window size has to be modulated accordingly when the modality varies. To enhance transferability, we propose a well-designed signal processing pipeline that preserves information in both the frequency and time domains across multiple scales. We begin by calculating the first and second derivatives for each single signal series, as suggested by Slapničar

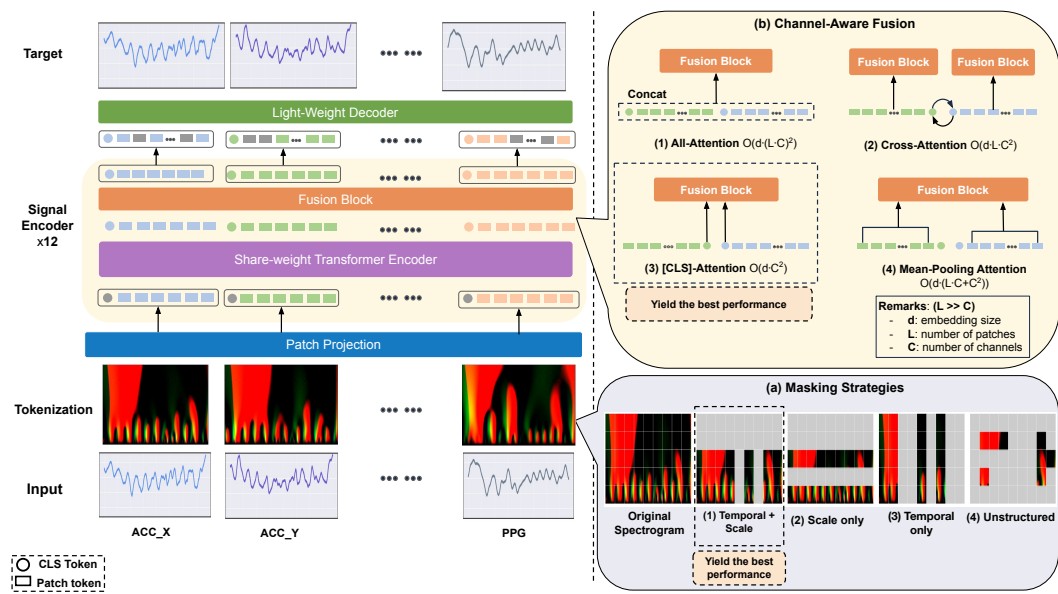

Figure 2: Overview of the pretrain pipeline.

et al. (2019), followed by computing the continuous wavelet transform (CWT) on both the raw and derivative series, resulting in three scalograms. Then, we stack the three scalograms to form data in RGB-image-like format. The derivatives capture the rate of signal change at different moments, while the wavelet transform provides a multi-resolution encoding that preserves information from both the time and frequency domains Torrence & Compo (1998). For the wavelet transform, we use the Mexican Hat wavelet for signal convolution, as recommended by previous studies (Burke & Nasor, 2004; Hosni & Atef, 2023; Hassani, 2021; Negi et al., 2024; Nedorubova et al., 2021b). We apply scales ranging from 1 to 64, following the guidance of (Sengupta et al., 2022; Nedorubova et al., 2021a), which sufficiently covers most frequency bands of interest for physiological signals. Finally, this RGB-like scalogram is divided into patches, which is treated in the same way as tokens in an ViT (Dosovitskiy et al., 2020). In this way, this tokenization approach can be applied to various types of sensing signals without sensor-specific adjustments or reconfigurations.

## 2.3 Model architecture and pre-train strategies

Following the tokenization step, we adopt common reconstruction-based pretraining strategies from Masked Auto Encoder (MAE) (He et al., 2021; Huang et al., 2023; Zhang et al., 2023a), which applying masking to input tokens and and optimizing the model using mean squared error (MSE) for reconstructing the raw time series. Inspired by Huang et al. (2023), we experiment with four masking strategies, as shown in Figure 2 (a), including masking on (1) temporal and scale, (2) scale only, (3) temporal only, and (4) unstructured axes. We observe that the temporal and scalar masking yields the best performance for the downstream tasks.

For the model architecture, we construct the backbone of our proposed framework with a convolutional patching layer followed by 12 standard Transformer blocks (Vaswani et al., 2023). For the same reason, NORMWEAR uses a lightweight decoder consisting of 2 Transformer blocks, combined with a linear projection layer and a convolution layer to reconstruct the raw physiological signals both temporally and spatially. We also prepend a special token [CLS] at each signal channel, aiming to learn and extract a generic representation for each signal.

Another important point to consider is that although empirical studies (Nie et al., 2023; Abbaspourazad et al., 2023) show that channel-independent structures effectively capture local patterns, they fail to account for relationships across channels. To address this, we introduce a channel-aware attention (fusion) layer after every other encoder block to incorporate cross-channel information. We explore several fusion approaches as shown in Figure 2 (b), with each method described below:

(1) **All-Attention Fusion:** This approach involves concatenating all tokens from each modality without considering their individual properties and fusing the information through a self-attention

module. However, this method requires quadratic computation time, as every token passes through the self-attention module, making it impractical for real-world applications.

(2) **Cross-Attention Fusion:** In addition to the cross-attention mechanism used in Cross-ViT (Chen et al., 2021), we introduce a slight modification to fit in our problem setting. We propose a symmetric fusion method, using the [CLS] token from each modality as an intermediary to exchange information between the patch tokens of another modality, then projecting the information back to its original modality in the subsequent Transformer layer. While this strategy is efficient, it restricts the model to handling only two time series signals or modalities, which deviates from our goal of building a general model capable of processing an arbitrary number of channels.

(3) **[CLS]-Attention Fusion** The [CLS] token serves as an abstract global representation for each signal modality. Here, we propose a hybrid fusion approach. We stack the [CLS] tokens from all signal modalities and perform feature fusion using a self-attention mechanism. The fused [CLS] token is then reattached to its original channel, enabling the newly learned information to be propagated to each patch token in subsequent transformer encoder layers.

(4) **Mean-Pooling Fusion** Similar to the [CLS]-Attention Fusion approach, we employ mean-pooling within each channel instead of using the [CLS] token as an abstract global representation.

Our empirical results show that [CLS]-attention fusion achieves the best performance for downstream tasks for our proposed NORMWEAR model. Details of all the ablation studies are reported in appendix D.

## 2.4 Zero shot inference with memory stream inspired fusion mechanism

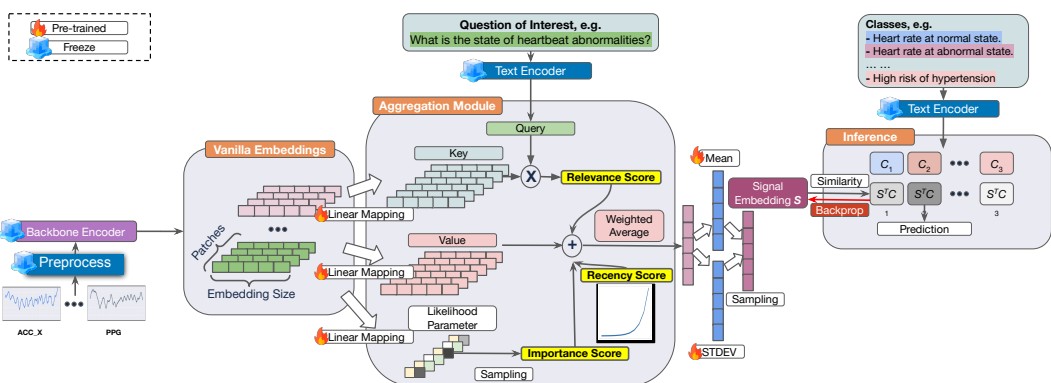

Figure 3: Memory stream inspired temporal fusion mechanism for representation alignment.

We enable zero-shot inference by introducing a novel temporal fusion mechanism that transforms multivariate sensing data into a unified representation within a text embedding space. Unlike prior approaches (Radford et al., 2021; Wu et al., 2023) that trained both signal encoder and text encoder jointly from scratch, our method is lightweight, as it does not require retraining these encoders.

For the objective of representation alignment specifically, with the semantic embedding of query sentence $q$ and backbone output $H \in \mathbb{R}^{P \times E}$ where $P$ is the patch size and $E$ is the embedding size, we will have the final fused representation $f(q, H) = \hat{Y} \in \mathbb{R}^E$ which the fusion function $f$ will be described in details in the following subsections. We then leverage the semantic embedding of ground truth sentence $Y$ to supervise the fused output $\hat{Y}$ with integrated loss function with penalty on Manhattan distance and cosine similarity, aiming to align the physiological representation with the same direction and magnitude as the semantic representation:

$$Loss(Y, \hat{Y}) = \lambda|Y - \hat{Y}| + \left(1 - \frac{Y \cdot \hat{Y}}{\|Y\|\|\hat{Y}\|}\right) \tag{1}$$

where $\lambda$ is hyper-parameters controlling the weight of loss components. During pretraining on the pretraining datasets stated in Table 2, we introduce both classification and regression tasks, as well as data augmentation with multiple alternative sentence patterns for each paired datasets, in order to allow the model to have a better estimation of the representation transformation function from the physiological signal space to the semantic space.

In this method, we leverage text as a common modality, mapping input signals into a unified textual space. By inferring within this shared space, we can assess the similarity between aligned physiological representations and potential ground-truth states, enabling zero-shot inference. However, relying solely on the cross-attention (relevance) score for temporal fusion is insufficient for human sensing tasks, as it overlooks temporal proximity, the contextual importance of each patch, and the intrinsic variations within each representation. Human sensing tasks, such as gesture recognition or physiological monitoring, often require prioritizing recent temporal patterns due to their stronger correlation with immediate human actions or conditions (Chowdhury et al., 2020; Chaudhury et al., 2021). To this end, we introduce recency scores, which assign higher weights to patches closer to the most recent time step in the sequence. Additionally, during vector aggregation, we adopt a variational-inspired approach (Kingma & Welling, 2022) where we compute the mean and standard deviation of patch embeddings before sampling. This design injects stochasticity into the representation, encouraging the model to explore and capture nuanced variations in human sensing data.

**Memory stream inspired fusion mechanism (MSiTF).** As mentioned above, the NormWear encoder have latent output shape of $H \in \mathbb{R}^{P \times E}$. Such an embedding vectors of all the patches have to be aggregated (average pooling by default) to form a fixed length representation suitable for non-sequential downstream tasks including classification or regression. Inspired by the philosophy of memory stream retrieval from the design of virtual game characters in Park et al. (2023), we implemented a novel fusion mechanism named MSiTF to generate representations optimized for human sensing, shown in Figure 3. Intuitively, MSiTF fuses the latent representations from all time steps before the final output layer with weighted scores computed according to **(1)** *how relevant they are to the objective tasks*, **(2)** *how important they are to the data itself*, and **(3)** *how close they are to the most current time step*. The output layer is instructed to select the most informative representations to optimize the objective task of representation alignment.

As outlined in Figure 3, we consider the *Relevance* score to be the cross-attention score between the sentence embedding generated by the pretrained language model (Muzammil, 2021) of the query sentence and the key representation of the embedding of each time step. For the *Recency* score, we use an exponential decay function, where the further the time step to the most recent time step, the lower the score. Finally, we consider the importance score IMP in this case to be whether to keep the representation at each time step or not. In order to achieve this, we assign binary parameters to each time step, denoted as $\theta_t = p(v_t) \in \mathbb{R}^2$ where $v_t \in \mathbb{R}^E$ is the representation vector at time step $t$ and $p$ is a trainable linear transformation function which will be optimized during pretraining. We then have the importance score for each patch defined as

$$W_{imp}(t) = \underset{i \in \{0,1\}}{\arg\max} \frac{\exp\left(\left(\log(\theta_{t,i}) + \epsilon\right)/\tau\right)}{\sum_{j \in \{0,1\}} \exp\left(\left(\log(\theta_{t,j}) + \epsilon_j\right)/\tau\right)} \tag{2}$$

where $\epsilon$ is the noise term sampled from Gumbel distribution (Jang et al., 2017), and $\tau$ is the temperature controlling the sharpness of the softmax function. Because $\arg\max$ is not a differentiable function, we will directly take the resulting probability corresponding to index at $j = 1$ to be the *importance* score, with $\tau$ being set to a small number to push the result closer to one hot vector from the softmax function. As a result, the trainable linear transformation will be optimized to determine whether to activate the gate during forward pass on each input signals. The final score for each patch is the summation of the three scores as described above. This score will be treated as the weight for aggregating the representations from all the patches to form the fixed length embedded output (vector with size of 768 in our case). This aggregated vector is then passed to the successive tasks on representation alignment and downstream task inference.

## 3 Experiments

In this section, we present a comprehensive evaluation across 11 publicly available datasets, focusing on 18 widely-recognized digital healthcare tasks. We first assess the transferability advantage of our proposed model compared to the solid baselines. Additionally, we examine the zero-shot capabilities of *NormWear*. Finally, we conduct nonlinear dynamics analysis on the waveform features across intermediate encoder layer to inspect model's behaviors.

### 3.1 Selection of baselines covering representative modeling strategies

Modeling multivariate wearable signals with arbitrary input channels and sensor types, such as those capturing activities of heart, brain, and body physical motions, presents unique challenges, as no

universally recognized open-source baseline or state-of-the-art (SoTA) model exists in this domain. To evaluate our approach, we selected diverse and representative baselines (as shown in Table 2).

In the literature, different modeling strategies have been proposed. Firstly, early approaches involved handcrafting statistical features, which was a widely adopted practice in signal processing (Yan et al., 2023a; Reyes-Ortiz et al., 2012; Mikelsons et al., 2017). We include this simple baseline as sanity check. Secondly, since sensory data can be naturally represented as time series (Woo et al., 2024; Semenoglou et al., 2023), we benchmarked our model against Chronos (Ansari et al., 2024) , as well as the common self-supervised framework TF-C (Zhang et al., 2022). Finally, the spectrum-based modeling methods (Vishnupriya & Meenakshi, 2018; Chun et al., 2016; Krishnan et al., 2020) are widely used for signal modeling. Therefore, we incorporate CLAP (Wu et al., 2023) into baselines that has demonstrates SoTA performance in spectrogram-based modeling. These baselines span distinct paradigms, providing a solid foundation to demonstrate the strengths of our model in wearable signal tasks.

### 3.2 Downstream evaluation, NormWear achieves the peak performance

We perform supervised training to evaluate the representation with linear probing on each downstream dataset. Performance is then assessed in the test set of these datasets. The classification tasks are solved by Newton's method with conjugate gradient, with AUC ROC being reported as main metric. The regression (vital signs) tasks are solved by Cholesky's method with closed form solution, with relative accuracy being reported. All scores are the higher the better.

From Figure 5, Table 3, and Table 4, we observe that NormWear consistently achieves peak performance across all task groups, including activity recognition, EEG signal analysis, disease risk evaluation, and vital sign estimation. Furthermore, its leading performance remains consistent across various evaluation metrics. Based on the macro-averaged total score across task groups, NormWear delivers a 3.6% improvement over the state-of-the-art (SoTA) time-series self-supervised learning framework, a 5.3% improvement over the SoTA spectrum-based modeling method, a 5.6% improvement over SoTA time-series forecasting models with LLM backbones, and a 5.3% improvement over standard statistical baselines. On larger datasets, NormWear significantly outperforms the statistical baseline by 9.0% and 7.5% for activity recognition and EEG brain activity monitoring tasks, respectively. On smaller datasets, it still achieves peak performance in disease risk evaluation. For vital sign estimation, all methods yield comparable results, suggesting inherent challenges in these regression tasks that warrant further investigation but are beyond the scope of this study.

These findings illustrate NormWear's capacity to balance consistency and adaptability across a diverse range of tasks and conditions. By excelling across standard benchmarks while addressing the intricacies of varied applications, NormWear exemplifies the philosophy of a foundation model: a reliable generalist capable of performing robustly across both typical and challenging scenarios.

### 3.3 Scaling up the Pretraining Data Size

In addition to demonstrating that NormWear outperforms all strong baselines, we further investigate the effect of varying pretraining data size on the model's downstream performance to examine whether the scaling law applies to our proposed methodology. As shown in Figure 4, the overall performance (measured by accuracy) significantly improves as the pretraining data size increases from approximately 37k (62 hours) to nearly 2.5M (4000 hours) samples of wearable signal data. This observation indicates that our model adheres to the scaling law, highlighting its potential scalability and suitability for future large-scale applications.

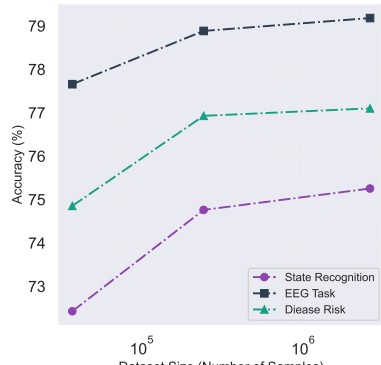

Figure 4: Scaling on downstream tasks.

### 3.4 The first zero-shot enabled foundation model for wearable sensing health applications

We achieve zero-shot inference by pretraining our proposed novel temporal fusion module on the task of representation alignment following the guidance in (Zhang et al., 2024a; Liu et al., 2024) to map the embedding from our proposed foundation model to semantic space. During test-time inference on downstream datasets, each ground truth label is converted into a sentence (details in appendix. B), which is transformed into a text embedding using a frozen text encoder. The sentence

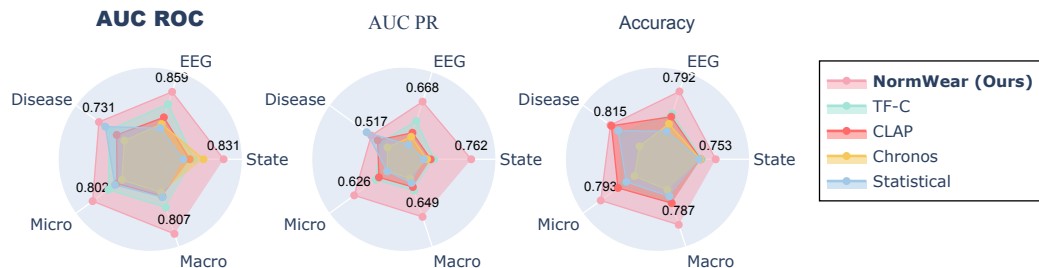

Figure 5: Overview of performance trend of NormWear against competitive baselines in downstream tasks: **(1) Disease** risk predictions. **(2) EEG** main tasks (mental and abnormal states prediction). **(3) State** recognition: physical and mental activities. **(4) Macro**: Average performance over types of tasks. **(5) Micro**: Average performance over each task.

Table 3: Performance on various downstream wearable-signal-based health related applications under linear probing evaluation.

| Downstream Tasks | Statistical | Chronos | CLAP | TF-C | NormWear (Ours) |
|---|---|---|---|---|---|
| WESAD | 66.213 | 71.489 | 72.383 | 69.865 | **76.060** |
| UCI-HAR | 95.784 | 91.593 | 96.420 | 96.892 | **98.954** |
| DriverFatigue | 63.249 | **76.722** | 61.889 | 66.882 | 74.292 |
| **Activity Recognition Avg.** | 75.082 | 79.935 | 76.897 | 77.880 | **83.102** |
| Epilepsy (eye open) | 82.489 | 82.41 | 85.094 | 89.153 | **92.743** |
| Epilepsy (eye close) | 87.457 | 88.218 | 89.867 | 94.416 | 94.828 |
| Epilepsy (health area) | 86.274 | 81.08 | 83.711 | 85.619 | **88.541** |
| Epilepsy (tumor area) | 82.816 | 81.034 | 83.644 | 86.348 | 87.197 |
| Epilepsy (seizure) | 88.272 | 97.572 | **97.734** | 93.998 | 97.053 |
| GAMEEMO | 51.009 | 53.747 | 52.551 | **56.275** | 54.937 |
| **EEG Main Tasks Avg.** | 79.720 | 80.677 | 82.100 | 84.302 | **85.883** |
| ECG-Abnormal | 97.092 | 98.585 | 97.23 | 98.275 | **99.140** |
| PPG-BP (HTN) | 59.499 | 52.425 | 56.757 | **65.229** | 62.341 |
| PPG-BP (DM) | 47.823 | 51.164 | 42.455 | **57.883** | 55.893 |
| PPG-BP (CVA) | **71.25** | 50.278 | 51.667 | 58.125 | 70.625 |
| PPG-BP (CVD) | 51.219 | 58.31 | 50.91 | **58.674** | 51.773 |
| PhysioNet EMG | **99.309** | 61.6 | 98.627 | 78.308 | 99.216 |
| **Risk Evaluation Avg.** | 71.032 | 62.060 | 66.274 | 69.416 | **73.165** |
| Noninvasive-BP | 92.31 | 91.79 | 91.922 | 87.481 | **92.420** |
| PPG-Hgb | 94.219 | **95.005** | 94.291 | 93.408 | 94.632 |
| Fetal-fPCG | 98.929 | 99.048 | **99.195** | 99.077 | 99.072 |
| **Vital Signs Avg.** | 95.153 | 95.281 | 95.136 | 93.322 | **95.375** |
| **Micro Avg.** | 78.623 | 76.782 | 78.130 | 79.773 | **82.762** |
| **Macro Avg.** | 80.247 | 79.488 | 80.103 | 81.230 | **84.381** |

with the closest distance with the embedding from our foundation model is used as the final inferential result. We also include the SoTA spectral-based model CLAP Wu et al. (2023) as a baseline to provide a more comprehensive comparison of the results. For CLAP, we experimented with both Manhattan distance (MD) and dot product (DP) as similarity metrics during inference. From table 5, we could observe that overall, the models equipped with our temporal proposed novel fusion mechanism outperform the baselines including leveraging the vanilla attention fusion mechanism. Although the final performance may not surpass that of linear probing, our work offers a significant contribution as an initial attempt to enable zero-shot inference through a lightweight pipeline across various wearable sensing healthcare tasks, without the need to rely on existing generative language models. We present this outcome to demonstrate that, even without fine-tuning, the model is capable of learning informative representations that can be directly leveraged for downstream tasks. Furthermore, as shown in Section 3.2, even a straightforward adaptation, such as linear probing, can yield notably improved results.

Table 4: [Updated] Details of Incidental Performance Metrics.

| Task Group | Methods | AUC ROC | AUC PR | Accuracy | Precision | Recall | F1 Score |
|---|---|---|---|---|---|---|---|
| Activity Recognition | Statistical | 75.082 | 63.996 | 65.298 | 61.450 | 61.56 | 61.034 |
| | Chronos | 79.935 | 65.622 | 66.175 | 62.044 | 61.512 | 60.522 |
| | CLAP | 76.897 | 67.026 | 66.349 | 62.790 | 62.826 | 62.435 |
| | TF-C | 77.880 | 68.228 | 67.175 | 64.967 | 64.798 | 64.783 |
| | NormWear (Ours) | **83.102** | **76.232** | **75.254** | **72.606** | **72.177** | **72.053** |
| EEG Main Tasks | Statistical | 79.720 | 50.172 | 73.921 | 63.567 | 57.529 | 57.948 |
| | Chronos | 80.677 | 55.507 | 75.285 | 72.442 | 52.520 | 47.671 |
| | CLAP | 82.100 | 57.518 | 76.391 | 68.506 | 61.961 | 62.650 |
| | TF-C | 84.302 | 61.864 | 76.825 | 71.702 | 65.517 | 67.889 |
| | NormWear (Ours) | **85.883** | **66.841** | **79.182** | **72.485** | 69.158 | 69.698 |
| Disease Risk Evaluation | Statistical | 71.032 | **53.783** | 79.688 | 52.718 | 53.235 | 50.807 |
| | Chronos | 62.060 | 40.673 | 71.910 | 45.512 | 43.739 | 40.569 |
| | CLAP | 66.274 | 48.232 | 81.327 | 53.028 | 54.721 | 52.804 |
| | TF-C | 69.416 | 46.312 | 78.929 | 52.123 | 52.352 | 51.349 |
| | NormWear (Ours) | **73.165** | 51.666 | **81.530** | **54.133** | **56.314** | **54.428** |
| Micro Average | Statistical | 75.317 | 51.596 | 74.503 | 58.804 | 56.618 | 55.709 |
| | Chronos | 73.082 | 51.596 | 72.113 | 59.590 | 50.806 | 47.401 |
| | CLAP | 74.729 | 55.705 | 76.357 | 61.171 | 59.238 | 58.669 |
| | TF-C | 77.063 | 56.916 | 75.737 | 62.523 | 60.107 | 60.652 |
| | NormWear (Ours) | **80.240** | **62.649** | **79.336** | **65.168** | **64.624** | **64.061** |
| Macro Average | Statistical | 75.278 | 55.983 | 72.969 | 59.245 | 57.441 | 56.596 |
| | Chronos | 74.224 | 53.934 | 71.123 | 59.999 | 52.590 | 49.587 |
| | CLAP | 75.091 | 57.592 | 74.689 | 61.441 | 59.836 | 59.296 |
| | TF-C | 77.199 | 58.801 | 74.310 | 62.931 | 60.889 | 61.340 |
| | NormWear (Ours) | **80.717** | **64.913** | **78.656** | **66.408** | **65.883** | **65.393** |

Table 5: Zero-shot performance on the downstream datasets, with AUC ROC being reported. The last two columns show the average across the tasks and across group types respectively.

| Model | WESAD | UCI-HAR | DriverFatigue | GAMEEMO | Epilepsy (eye open) | Epilepsy (eye close) | Epilepsy (health area) | Epilepsy (tumor area) | Epilepsy (seizure) | PPG-BP (HTN) | PPG-BP (DM) | PPG-BP (CVA) | PPG-BP (CVD) | ECG-Abnormal | PhysioNet EMG | Micro Avg. | Macro Avg. |
|---|---|---|---|---|---|---|---|---|---|---|---|---|---|---|---|---|---|
| CLAP - MD | 45.3 | 62.8 | 58.5 | **53.1** | 44.9 | 45.1 | 47.6 | 30.5 | **84.9** | **59.4** | 41.8 | 46.0 | 57.4 | 22.9 | 55.4 | 50.4 | 51.2 |
| CLAP - DP | 50.7 | 52.3 | **61.1** | 51.6 | **54.4** | 41.9 | 58.6 | 46.4 | 74.3 | 52.2 | 41.4 | 50.6 | 58.9 | 42.7 | 38.3 | 51.7 | 52.2 |
| **NORMWEAR** w/ MSiTF | 55.9 | **71.4** | 54.9 | 50.2 | 54.0 | 56.4 | **66.9** | **57.4** | 53.7 | 56.5 | 53.2 | **65.0** | **63.1** | **74.3** | **65.7** | **59.9** | **60.1** |
| - w/o IMP | **56.2** | 70.3 | 55.4 | 49.8 | 54.0 | **56.5** | **66.9** | 57.3 | 52.9 | 56.5 | **54.3** | 61.7 | 60.7 | 73.4 | 65.2 | 59.4 | 59.6 |
| - w/o text aug | 54.8 | 65.8 | 55.2 | 49.2 | 31.0 | 58.4 | 58.6 | 32.8 | 58.1 | 50.2 | 52.6 | 50.8 | 50.6 | 47.7 | 33.6 | 50.0 | 51.4 |

## 3.5 Quantify the observed intrinsic behaviors: nonlinear dynamics analysis on the features from each layer

Understanding the representations extracted by intermediate layers is crucial to interpreting our model's behavior. To quantify the meaningfulness of these representations, we conducted a nonlinear dynamics analysis inspired by chaos theory. This method analyzes the features' intrinsic behaviors through metrics like the Lyapunov exponent (Wolf et al., 1985) (sensitivity to initial conditions), Hurst exponent (Qian & Rasheed, 2004) (self-correlation/seasonality), and persistence entropy (Yan et al., 2023b) (unpredictability in system states). We obtain the following key observations:

**1. Deeper Layers Capture Higher-Order Complexity.**
- For signals such as GSR, EEG, and ACC, deeper layers show lower self-correlation (DFA (Hu et al., 2001)) and higher unpredictability (persistence entropy), indicating a transition to representations that are less periodic and more chaotic.

- The decrease in the Lyapunov exponent across layers suggests reduced variation in extracted features, aligning with the idea that deeper layers capture more abstract, long-term patterns with broader receptive fields.

**2. Modalities with Simpler Dynamics.** In contrast, PPG and ECG signals, dominated by regular heart activity, exhibit more stable patterns across layers. This aligns with their simpler waveform structures and less complex dynamics compared to signals related to neural and physical activities.

These visualizations reveal that the model progressively transforms raw sensory data into representations aligned with the complexity of each signal. For GSR and EEG, deeper layers exhibit increased unpredictability and reduced periodicity, highlighting the extraction of nuanced, higher-order patterns critical for human sensing. In contrast, the stability of representations for PPG and ECG reflects their simpler dynamics, demonstrating the model's adaptability to varying signal characteristics. This analysis confirms that the intermediate representations are purposefully optimized to capture the temporal and structural nuances of each modality, supporting the conclusion that the model learns meaningful features tailored to human sensing tasks.

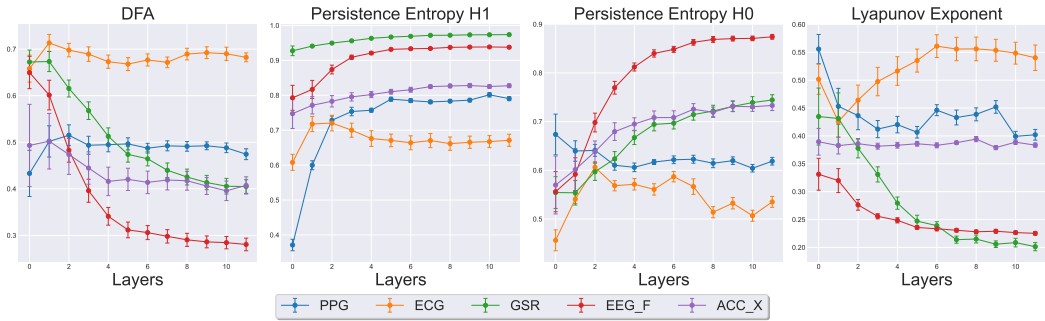

Figure 6: Nonlinear dynamic analysis on the waveforms extract at different layers of our model.

## 4 LIMITATIONS AND CONCLUSION

In this work, we mainly propose a foundation model for wearable physiological signals. There are three main limitations. Firstly, for the representation alignment pipeline, although we make significant efforts to augment the text data and add a variational sampling mechanism, we have a relatively limited set of wearable sensing-based healthcare objectives during pretraining. Drawing insights from the natural language processing domain (Devlin et al., 2019), we encourage future studies to increase the diversity of tasks for pretraining to achieve a more promising performance. Secondly, regarding zero-shot inference, the current pipeline design of NORMWEAR is more aligned with a classification scenario, also as suggested in Wu et al. (2023) and Zhang et al. (2024b). The most straightforward approach for regression would be to discretize the target label into bins. However, this approach does not fully address the challenge of adapting to regression tasks. Therefore, we recommend exploring alternative modeling strategies for performing zero-shot learning on continuous scales. Finally, human sensing includes signals from a wide range of frequency bands. For example, audio data, as one of the popular modalities in contactless sensing, has a much higher and wider range of frequencies of interest. In contrast, lower-frequency data are more common in clinical research. For instance, most wearable devices record only minute-to-minute data such as heart rate, estimated calories consumed, and noise level around. Medical-related bio-markers are day-to-day data such as measurements of glucose level, blood pressure, and estimated body fat. In the current design, NORMWEAR does not incorporate such a wide variety of frequency ranges; however, there is great potential to verify and improve its ability when extending to other type of signals with a wider range of frequency bands of interest, which is a future research scope.

In conclusion, NORMWEAR is a practical tool that could serve as a starting point for researchers and clinicians when tackling a problem with wearable sensing based signal data. Our proposed model could extract informative embedding representations from raw signal series, which can be leveraged for further machine learning modeling, clustering, embedding vector-based information retrieval, and deployment of real-time health states monitoring with minimal tuning. We've justified the utilizability and generalization of NORMWEAR through an extensive evaluation of various ubiquitous health applications. Along with the interpretability analysis, our work could provide a transparent understanding of the model's inner feature extraction and importance assignment processing. As for future works, it is important to leverage our proposed model on more large scale clinical applications and explore the applicability of embedding vectors as state representations for intervention modeling problems that comprise the decision-making process. We also suggest extending the proposed model on contactless sensing signals, as mentioned previously, such as audio and thermal imaging, which could provide more thorough health-related information.

ETHICS STATEMENT

This study contains applications in the field of healthcare. We ensured that all the data being used during pretraining and evaluations were made publicly available by the original authors, and all these works were cited properly.

REPRODUCIBILITY STATEMENT

The full code base is submitted in supplementary material referred to as *normwear_codebase.zip*, comprising all the scripts for exploratory data analysis and preprocessing, model construction, pretraining, downstream evaluation, result analysis, and all the visualizations that are described in this paper. The GitHub repository containing all the documentation will be published simultaneously with the paper.

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

APPENDIX

## A  RELATED WORK

Self-supervised learning paradigm, coupled with large and diverse datasets, has gained popularity recently due to its adaptability to various downstream tasks (Bommasani et al., 2022). This approach has attracted significant interest in the wearable sensor domain, particularly for applications in physiological signal analysis. Recent studies have utilized self-supervised learning in wearable devices for tasks such as activity recognition (Spathis et al., 2021; Yuan et al., 2023; Zhang et al., 2023b). Additionally, it has been applied to physiological signals such as PPG, ECG, and EEG, spanning various healthcare monitoring tasks (Abbaspourazad et al., 2023; Pillai et al., 2024; Zhang et al., 2019; Mehari & Strodthoff, 2022; Mohsenvand et al., 2020). However, these studies often rely on a predefined set of devices, which limits the models' adaptability when clinical application settings change. For instance, when new devices or modalities are introduced, these models, which have not been exposed to such data during training, often require fine-tuning to remain functional. Furthermore, many of these models are not publicly accessible due to the sensitivity of healthcare data, which hinders progress in this area. These challenges underscore the need for an open, pre-trained model that can accommodate various device configurations and adapt to evolving clinical requirements.

On the other hand, general time series models (Ansari et al., 2024; Woo et al., 2024; Zhang et al., 2022) have shown significant advancements but are predominantly trained and evaluated in domains such as transportation, energy consumption, and finance, with limited exploration in physiological signals. While physiological signals are inherently multivariate time series, these models have not been trained on such data, leaving their transferability to the physiological sensor domain uncertain. Ignoring the correlations between different sensors may result in suboptimal performance when applied to this domain. Motivated by these limitations, this work focuses on developing a foundation model for sensory time series data capable of accommodating arbitrary combinations of device modalities as multivariate series, while investigating strategies to effectively leverage sensor correlations.

## B  IMPLEMENTATION DETAIL

**Data Preprocess.** For the data preparation, we set the uniform sampling rate and interval length to 65 HZ and 6 seconds respectively. In our case, 65 Hz covers most of the frequency bands of interest such as heart activity, physical motions, and neuron activity up to the beginning of Gamma power (above 30 Hz). And a great amount of samples are less than 6 seconds such as (Reyes-Ortiz et al., 2012; Liang et al., 2018; Bousseljot et al., 2009). We conduct basic pre-processing for each signal with identical setting: (1) de-trended by subtract the result of a linear least-squares fit to series data from the raw time series, and (2) Gaussian smoothed with standard deviation of 1.3 (0.02 seconds), ensuring a highly consistent dataset for training.

Since the Transformer's computational requirements scale quadratically with input length, to release the full potential of our self-supervised algorithm, we segment our multivariate time series into intervals with a uniform length and pad shorter samples with zeros. This approach not only enables parallel processing of samples in large minibatches but also addresses variation in the length of individual samples.

For the downstream task, we split the data into train and test sets for linear probing evaluation with portion of $80\%$ and $20\%$ correspondingly. The split is stratified on the anonymized subject ID if this information is provided by the dataset.

**Data Augmentation.** Since there are very few publicly available datasets containing multiple devices or modalities, we aim to expand our curated training set to fully leverage the potential of self-supervised learning. Inspired by data augmentation techniques in computer vision and natural language processing (Zhang et al., 2017; Carmona et al., 2021), we adopt a heuristic approach to augment the dataset. Specifically, we augment each sub-dataset by a factor of 10. For each dataset, we sample two time series, randomly extract a segment from one, and substitute it with a trans-

formed counterpart, as outlined in the pseudocode in Algorithm 1. As a result, our training set is expanded to 2,586,404 segments, corresponding to 4,310 hours of data.

---

**Algorithm 1** Time Series Mixup Augmentation

---

**Input:** Time series dataset $\mathcal{X}$, number of augmentations $n$
**Output:** Augmented Dataset $\tilde{\mathcal{X}}$

1: **for** $i = 1$ to $n$ **do**
2:     Sample two time series $\mathbf{x}^{(1)}, \mathbf{x}^{(2)} \sim \mathcal{X}$
3:     Sample a chunk size $\lambda \sim \mathcal{U}(0, l)$
4:     Sample start indices $s_1, s_2 \sim \mathcal{U}(0, l - \lambda)$
5:     Swap chunk from $\mathbf{x}^{(2)}$ into $\mathbf{x}^{(1)}$:

$$\mathbf{x}^{(1)}_{s_1:s_1+\lambda} \leftarrow \mathbf{x}^{(2)}_{s_2:s_2+\lambda}$$

6:     Append $\mathbf{x}^{(1)}$ into $\tilde{\mathcal{X}}$
7: **end for**
8: **return** $\tilde{\mathcal{X}}$

---

**Pretraining Framework.** Normwear is derived from the Masked Autoencoder (MAE) (He et al., 2021). The detailed hyper-parameter choice is descibe in 6. We use a Conv2D layer with a kernel size of (9, 5) and a stride of (9, 5), ensuring no overlapping patches. This layer takes input with 3 channels and projects it to 768 channels, matching the hidden size of our encoders. In Normwear, we apply structured masking independently to each variate along both the frequency and time axes, with respective masking ratios of 0.6 and 0.5. This results in an expected overall masking ratio of 0.8 for each variate. Only the unmasked tokens are passed to the encoder, reducing computational complexity. To enhance representation learning, we introduce six additional transformer blocks as fusion layers, interleaved with the original 12 encoder blocks, creating a total of 18 blocks. Each transformer block has a hidden dimension of 768 and uses LayerNorm as in the original MAE. The latent embeddings obtained from the encoder are projected from 768 to 512 dimensions. Learnable masked tokens are reinserted at their original positions, and positional embeddings are added to guide the decoder in reconstructing the input series. The lightweight decoder consists of two transformer blocks with a hidden dimension of 512, followed by two Conv1D layers. The first Conv1D layer maps from the flattened multivariate signal embedding to an intermediate dimension, and the second Conv1D layer maps from this intermediate dimension back to the original multivariate signal space. A GELU activation function is used between these layers, with BatchNorm applied to the input. The decoder reconstructs the original input series, and the model is trained using Mean Squared Error (MSE) loss on all data points. Our models are pre-trained for 45,000 steps with a batch size of 256, using the AdamW optimizer with a learning rate of $10^{-4}$. We did not perform on-the-fly data augmentation, as suggested in the MAE framework, due to the high masking ratio. (An end-to-end example of the input and output of this pretraining pipeline is illustrated in Fig. 7)

**MSiTF.** For pretraining the representation alignment module, we have the training hyper-parameters in Table 7.

**Sentence template example for signal-sext alignment.**
- Emotion Task:
  - 'The emotion detected is {}.',
  - 'This subject is feeling {}.',
  - 'The emotional state is {}.',
  - 'The identified emotion is {}.'

- Activity Task:
  - 'This subject is currently {}.',
  - 'The subject is engaged in {}.',
  - 'Current activity is {}.',
  - 'Subjectś activity is {}.'

where {} is the placeholder for the corresponding label of each sample in pretraining datasets.

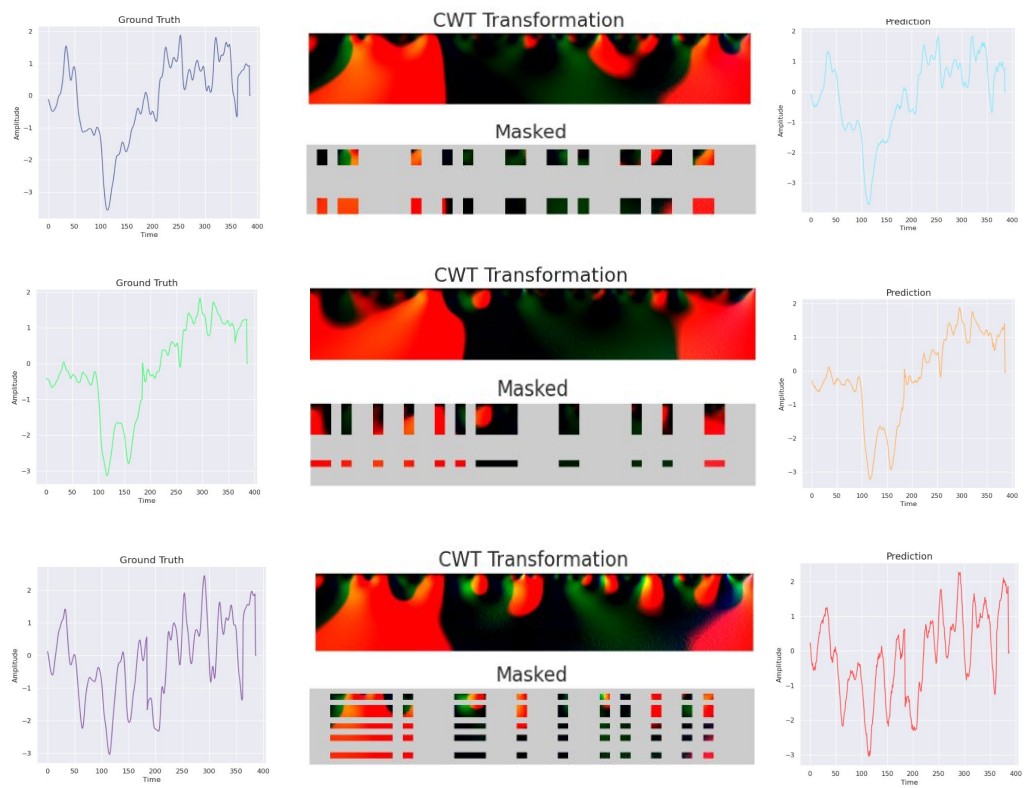

Figure 7: Visualization of original time series, CWT transformation image, masked image with structured masking, and reconstructed time series.

**Statistical Feature list:**

Features in *time domain*: mean, std, max, min, skew, kurtosis, 25% quantile, median, 75% quantile.

Features in *frequency domain*: centroid, spread, mean frequency, peak frequency, 25% quantile frequency, median frequency, 75% quantile frequency.

**Radar Plot or Performance Trend.** To enhance the visual contrast between model performances across tasks, we applied the Softmax function to the raw performance scores. This transformation rescales the scores to a range between 0 and 1, accentuating relative differences between models. While the Softmax transformation emphasizes the relative improvement of our model over others, we note that the absolute scores may differ from those originally reported.

## C    COMPLEXITY ANALYSIS OF DIFFERENT APPROACHES FOR CROSS-CHANNEL FUSION

When conducting multi-channel modeling, for example, when the input comprises an arbitrary number of signals, a fusion operation needs to be conducted across all channels in order to let the model extract correlation information. Because we will deploy the model on an edge device like Jetson Nano, other than empirical evidence of the performance, we also have to consider the computation complexity of different approaches. A brief visualization of the runtime complexity of different approaches is presented in figure 8. The detailed derivation is presented in the following subsections.

Table 6: NormWear Pretraining Hyper-parameters.

| Hyper-parameter | Value |
|---|---|
| # cross-patches Transformer Encoder | 12 |
| # cross-channels Transformer Encoder | 6 |
| # Transformer Decoder | 2 |
| # Attention Heads | 12 |
| Encoder Latent Size | 768 |
| Decoder Latent Size | 512 |
| Feedforward Latent Size | 3072 |
| Normalization | LayerNorm |
| Patch size (time axis) | 9 |
| Patch size (scale axis) | 5 |
| Optimizer | AdamW |
| Loss Scalar | NativeScaler |
| Base Learning Rate (blr) | 1e-3 |
| Epochs | 140 |
| Batch size | 192 |

Table 7: MSiFT Hyper-parameter

| Hyper-parameter | Value |
|---|---|
| Learning rate (lr) | 1e-3 |
| Epochs | 40 |
| Batch size | 32 |
| L2 regularization | 5e-6 |
| lr decay rate | 0.997 |
| $\lambda$ | 0.5 |
| $\tau$ | 0.5 |

## C.1 ALL-ATTENTION

For the approach of conducting self-attention by concatenating all the patches, we arrive the Big-O complexity expression as follows:

- We denote $C$ as the number of input channels, $d$ as the embedding size, $L$ as the number of patches convolved from the time series in each channel (proportional to sequence length), and $x \in \mathbb{R}^{C \times L \times d}$ as the input data before feeding into the fusion block. We have a total of $L \cdot C$ patches.

- When calculating the attention scores, dot products are computed for each pair of the patches, which results in the following calculation process:

  **for** $i$ in $[1, 2, ..., C]$ **do**
      **for** $j$ in $[1, 2, ..., L]$ **do**
          2) $N = \exp(\text{attn}(x_{i,j}))$, $\implies O(L \cdot C)$
          **for** $k$ in $[1, 2, ..., C]$ **do**
              **for** $l$ in $[1, 2, ..., L]$ **do**
                  1) Calculate dot product: $\text{attn}(x_{i,j}, x_{k,l}) = x_{i,j}^T x_{k,l}$, $\implies O(2d)$
                  2) Softmax over all-attention scores, $\frac{\exp(\text{attn}(x_{i,j}, x_{k,l}))}{N}$, $\implies O(1)$
                  3) Weighted Average: $x_{i,j} + \text{attn}(x_{i,j}, x_{k,l}) \cdot x_{k,l}$, $\implies O(2d)$
              **end for**
          **end for**
      **end for**
  **end for**

  where "1), 2), 3)" represents the operations conducted at the first, second, and third rounds of entering the entire nested loops. The complexity for the first round of operation results in a complexity of:

  $$\sum_{i=1}^{C}\sum_{j=1}^{L}\sum_{k=1}^{C}\sum_{l=1}^{L} 2d = \sum_{i=1}^{C}\sum_{j=1}^{L}\sum_{k=1}^{C} L \cdot 2d = \sum_{i=1}^{C}\sum_{j=1}^{L} C \cdot L \cdot 2d = O(d \cdot (L \cdot C)^2) \quad (3)$$

  where in the case of multi-head attention, the dot product still has the complexity of $O(2d)$, and because the number of heads is a constant, the final complexity is equivalent to the result in equation 3.

- Similarly, the softmax operation will result in a complexity of $O((L \cdot C)^2)$, and the final weighted average operation will also have a complexity of $O(d \cdot (L \cdot C)^2)$, which results in total complexity of

  $$O(d \cdot (L \cdot C)^2) + O((L \cdot C)^2) + O(d \cdot (L \cdot C)^2) = O(d \cdot (L \cdot C)^2) \quad (4)$$

## C.2 CROSS-ATTENTION

For the pairwise cross-attention approach following guidance of Chen et al. (2021), we have the operation defined as

> **for** $i$ in $[1, 2, ..., C-1]$ **do**
>     **for** $j$ in $[1, 2, ..., C]$ **do**
>         2) $N = \exp(\text{attn}(x_{i,1})), \implies O(L)$
>         **for** $k$ in $[2, 3, ..., L]$ **do**
>             1) Calculate $\text{attn}(x_{i,1}, x_{j,k}), \implies O(2d)$
>             2) Softmax over all-attention scores, $\frac{\exp(\text{attn}(x_{i,1}, x_{j,k}))}{N}, \implies O(1)$
>             3) Weighted average: $x_{i,1} + x_{j,k}, \implies O(2d)$
>         **end for**
>     **end for**
> **end for**

with the same notion in the previous subsection. The total complexity is

$$O(C^2 \cdot L \cdot 2d) + O(C^2 \cdot L) + O(C^2 \cdot L \cdot 2d) = O(d \cdot L \cdot C^2) \tag{5}$$

## C.3 [CLS]-ATTENTION

This is the approach that we adopted for the final version of our proposed foundation model. Only the embedding corresponding to the [CLS] token of each channel is involved during the self-attention operation. Therefore, the complexity is

$$O(d \cdot C^2) \tag{6}$$

## C.4 MEAN-POOL ATTENTION

For fusion with mean-pool attention, we first calculate the mean representation for each channel, resulting in a complexity of $O(C \cdot L \cdot d)$. And self-attention with Tese mean representations has the same complexity as [CLS]-attention, which is $O(d \cdot C^2)$. Thus, the total complexity is

$$O(C \cdot L \cdot d) + O(d \cdot C^2) = O(d \cdot (L \cdot C + C^2)) \tag{7}$$

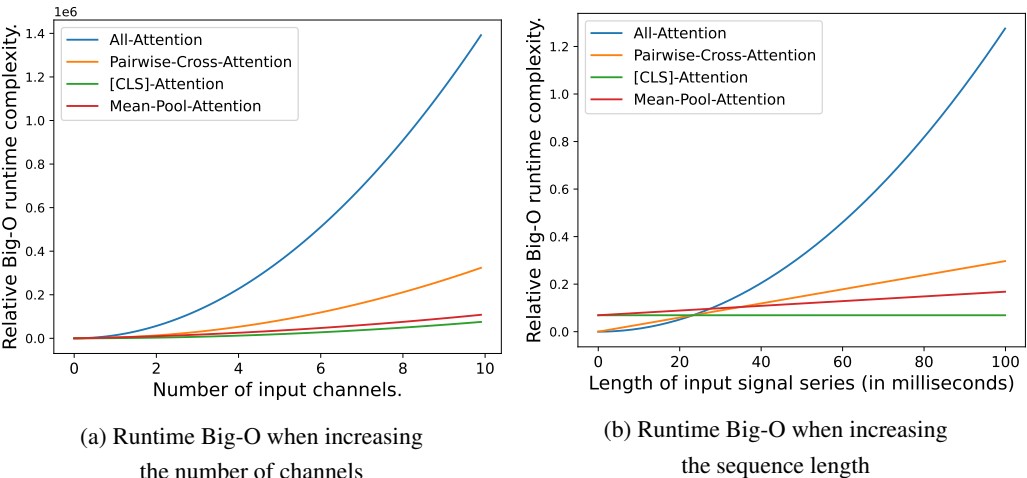

(a) Runtime Big-O when increasing the number of channels

(b) Runtime Big-O when increasing the sequence length

Figure 8: Visualization of runtime complexity when scaling up the number of channels or the sequence length.

# D  ABLATION STUDY

Due to computational constraints, we will conduct the ablation study on our smaller dataset (37k samples) to train and evaluate the model, establishing a proof of concept and demonstrating the effectiveness of our approach in a controlled setting.

**Fusion Schemes.** Table 8 shows the performance of different fusion schemes, including (1) no fusion, (2) cross-attention fusion, (3) [CLS]-attention fusion, and (4) mean-pooling fusion. We excluded all-attention fusion in our ablation study because it is computationally prohibitible. Among all the compared strategies, the [CLS] token fusion generally achieves the best accuracy with a minor increase in parameters.

Table 8: Performance Comparison of Various Fusion Schemes

| Dataset/Task | No-Fusion | All-Attention | Cross-Attention Fusion | [CLS] Token Fusion | Mean Pooling Fusion |
|---|---|---|---|---|---|
| Emotion Classification | 80.345 ± 0.022 | N/A | 76.231±1.069 | **80.373±0.049** | 79.010±0.030 |
| Valence-Arousal Prediction | 59.224 ± 0.732 | N/A | 60.271±0.522 | **62.386±0.175** | 60.930±0.240 |
| Driver Fatigue Detection | 68.086 ±0.040 | N/A | 69.241±0.167 | **71.009±0.080** | 70.230±0.350 |
| Human Activity Recognition | 95.320 ± 0.058 | N/A | 94.723±0.085 | **96.155±0.022** | 95.650±0.060 |
| Blood Pressure Estimation | **92.560±0.127** | N/A | 89.464±0.805 | 91.967±0.673 | 91.780±1.580 |
| Hemoglobin Estimation | 86.690±0.043 | N/A | 85.585±0.266 | 86.476±0.021 | **86.740 ± 0.010** |
| Fetal Heart Rate Estimation | 95.249±0.016 | N/A | 92.592±0.079 | **95.345±0.596** | 94.150±0.010 |
| Heartbeat abnormal Detection | 99.669±0.002 | N/A | 99.451±0.088 | 99.611±0.009 | **99.800 ± 0.010** |
| Hypertension Risk Evaluation | 64.460±1.071 | N/A | 63.065±0.751 | 66.896±0.276 | **67.520 ± 0.470** |
| Diabetes Risk Evaluation | 56.035±0.629 | N/A | 60.724±2.589 | **72.216±0.694** | 68.760±0.570 |
| Brain Stroke Risk Evaluation | 55.209±0.958 | N/A | **72.323±2.982** | 64.387±0.922 | 48.890±2.870 |
| Brain Disease Risk Evaluation | **64.969±1.964** | N/A | 55.532±1.568 | 55.897±1.057 | 63.510±6.060 |
| Average Score | 76.484 | N/A | 76.600 | **78.345** | 77.25 |

**Masking Strategies in Pre-training.** We ablated our masking strategy introduced in Section 2.3. Using a consistent mask ratio of 0.8 in all strategies, we found that applying masking along the scale and time axes produced the best performance (details in Table 9). **Input Representations.**

Table 9: Performance Comparison of Various Masking Strategies

| Dataset/Task | Unstructured ($P = 0.8$) | Structured ($P_t = 0.8, P_f = 0.0$) | Structured ($P_t = 0.0, P_f = 0.8$) | Structured ($P_t = 0.6, P_f = 0.5$) |
|---|---|---|---|---|
| Emotion Classification | 73.57 | 71.71 | 72.44 | 80.37 |
| Valence-Arousal Prediction | 59.44 | 61.77 | 61.32 | 62.39 |
| Driver Fatigue Detection | 64.90 | 74.38 | 75.54 | 71.01 |
| Human Activity Recognition | 95.40 | 95.45 | 95.40 | 96.20 |
| Blood Pressure Estimation | 91.78 | 88.58 | 91.99 | 91.97 |
| Hemoglobin Estimation | 88.56 | 88.97 | 88.78 | 86.48 |
| Fetal Heart Rate Estimation | 95.26 | 93.56 | 95.58 | 95.35 |
| Heartbeat abnormal Detection | 97.78 | 99.17 | 99.14 | 99.61 |
| Hypertension Risk Evaluation | 62.42 | 64.83 | 64.35 | 66.90 |
| Diabetes Risk Evaluation | 56.27 | 66.85 | 43.96 | 72.22 |
| Brain Stroke Risk Evaluation | 64.89 | 54.83 | 46.60 | 64.39 |
| Brain Disease Risk Evaluation | 45.18 | 61.74 | 51.80 | 55.90 |
| Average Score | 74.87 | 77.07 | 74.08 | **78.27** |

Table10 compares the performance of two input representations: (1) CWT scalogram and (2) raw time series. The CWT scalogram converts the time series into a time-frequency representation, while the raw time series retains the original sensor data. Among the two representations, the model trained on CWT scalograms demonstrates better performance, suggesting that the time-frequency features enhance model accuracy.

From Table 12, we observe that demographic information and represensations extracted from wearable signals have their own strength on different tasks, and most of the time, when we concatenate them together, the overall performance will be better. The performance drop in some cases after concatenation, which indicate that there might be some confounding relationship between these two

Table 10:  Performance Comparison Between CWT Scalogram and Raw Time Series as Inputs.

| Dataset/Task | Raw Series Input | CWT Scalogram Input |
|---|---|---|
| Emotion Classification | 73.60 | **80.37** |
| Valence-Arousal Prediction | 61.04 | **62.39** |
| Driver Fatigue Detection | **76.25** | 71.01 |
| Human Activity Recognition | **96.25** | 96.20 |
| Blood Pressure Estimation | 89.76 | **91.97** |
| Hemoglobin Estimation | 86.29 | **86.48** |
| Fetal Heart Rate Estimation | **95.88** | 95.35 |
| Heartbeat Abnormal Detection | 99.29 | **99.61** |
| Hypertension Risk Evaluation | 63.65 | **66.9** |
| Diabetes Risk Evaluation | 57.54 | **72.22** |
| Brain Stroke Risk Evaluation | 54.27 | **64.39** |
| Brain Disease Risk Evaluation | 51.13 | **55.90** |
| **Average Score** | 76.25 | **78.27** |

Table 11:  Performance on various downstream wearable-signal-based health related applications under linear probing evaluation using 5 fold cross validation stratified by subject ID (if provided by the data source). In this table, The classification tasks are solved by Newton's method with conjugate gradient, and the AUC ROC are reported. The regression (noninvasive BP estimate) tasks are solved by Cholesky's method with closed form solution for ridge regression, and the relative accuracy (1 minus relative error) are reported. All the scores are the higher the better.

| Downstream Tasks | Statistical | Chronos | CLAP | TF-C | NormWear-L (Ours) |
|---|---|---|---|---|---|
| WESAD | 79.992 +- 0.707 | 83.332 +- 0.841 | 87.824 +- 0.463 | 82.701 +- 0.536 | **89.585 +- 0.683** |
| UCI-HAR | 95.602 +- 0.148 | 91.956 +- 0.256 | 96.864 +- 0.175 | 97.382 +- 0.138 | **98.179 +- 0.06** |
| DriverFatigue | 69.614 +- 1.138 | **72.48 +- 2.848** | 66.251 +- 0.471 | 65.026 +- 1.198 | 68.971 +- 1.32 |
| GAMEEMO | 64.281 +- 1.292 | 56.694 +- 0.878 | 64.119 +- 0.543 | 62.925 +- 0.999 | **67.863 +- 0.72** |
| Noninvasive | 92.83 +- 0.386 | 92.223 +- 0.356 | 92.612 +- 0.272 | 88.707 +- 0.622 | **93.381 +- 0.516** |
| **Avg.** | 80.464 +- 0.734 | 79.337 +- 1.036 | 81.534 +- 0.385 | 79.348 +- 0.699 | **83.596 +- 0.660** |

Table 12:  Checking the reliance on demographic information.

| Downstream Tasks | Simple Baseline Mode and Mean | Demographic | NormWear-Medium | Demographic + NormWear-Medium | NormWear-Large | Demographic + NormWear-Large |
|---|---|---|---|---|---|---|
| WESAD | 50.000 | 49.907 | **74.227** | 69.06 | **76.06** | 68.755 |
| Noninvasive | 92.988 | 92.954 | 91.427 | 90.84 | 92.42 | 92.528 |
| PPG-Hgb | 94.816 | 95.634 | 94.911 | 95.835 | 94.632 | 96.384 |
| Fetal-fPCG | 99.033 | 99.039 | 98.997 | 99.001 | 99.072 | 99.097 |
| Vital Signs Avg. | 95.612 | **95.876** | 95.112 | 95.225 | 95.375 | **96.003** |
| PPG-BP (HTN) | 50.000 | 59.899 | 62.746 | 64.482 | 62.341 | 61.291 |
| PPG-BP (DM) | 50.000 | 47.297 | 62.613 | 47.86 | 55.893 | 60.135 |
| PPG-BP (CVA) | 50.000 | 81.875 | 67.639 | 83.681 | 70.625 | 77.847 |
| PPG-BP (CVD) | 50.000 | 71.011 | 51.504 | 70.37 | 51.773 | 67.466 |
| Risk Evaluation Avg. | 50.000 | 65.021 | 61.126 | **66.598** | 60.158 | **66.685** |
| Micro Avg. | 67.105 | 74.702 | 75.508 | **77.641** | 75.352 | **77.938** |
| Macro Avg. | 65.204 | 70.268 | 76.821 | **76.961** | 77.198 | 77.148 |

representations, hence further indicated that the information lies in demographic and the wearable represenation from NormWear are focused on different aspects. Same observation are observed with arbitrary model checkpoints during pretraining (denoted as Medium and Large marker representing different stage of training when we do the study on increasing the pretrain size.)

# E    DEPLOYMENT OF NORMWEAR: TESTING ON THE EDGE

As shown in the table 13, the GPU setup on an NVIDIA RTX 3090 significantly outperforms other configurations in inference speed, achieving an inference time of only 0.18 seconds while maintaining low RAM usage (8.04 MB) and moderate VRAM requirements (732.82 MB). In contrast, the CPU setup on MacOS M1 requires 4.21 seconds, reflecting a considerably slower performance despite similar RAM usage (9.12 MB) and no VRAM consumption. On edge devices, such as the Jetson Nano 4GB, the CPU-based setup exhibits the slowest inference time of 40.69 seconds, while the GPU variant improves this to 34.87 seconds with a VRAM requirement of 504.46 MB. Storage requirements remain constant across all configurations at 1.63 GB.

Table 13:   Computation resources consumed across various devices, on 6 channels data for 6 seconds.

| Dataset/Task | Infer time | RAM | VRAM | Storage |
|---|---|---|---|---|
| CPU (MacOS, M1) | 4.21 s | 9.12 MB | - | 1.63 GB |
| GPU - Debian GNU/Linux - NVIDIA-RTX-3090 | 0.18 s | 8.04 MB | 732.82 MB | 1.63 GB |
| Edge (Jetson Nano 4GB, CPU) | 40.69 s | 9.12 MB | - | 1.63 GB |
| Edge (Jetson Nano 4GB, GPU) | 34.87 s | 8.17 MB | 504.46 MB | 1.63 GB |

# F    [RE-POSITIONED] FEATURE VISUALIZATION

## F.1    THE MODEL IS AGNOSTIC TO THE INPUT SIGNALS

This section investigates whether, without requiring the signal modality type information as input, NORMWEAR can effectively distinguish between different signal sources. We randomly sampled 500 samples for each sensor type and fed them into our pretrained model. We use t-SNE (Van der Maaten & Hinton, 2008), with PCA (Jolliffe & Cadima, 2016) initialization to visualize the learned representations corresponding to the [CLS] special token at the last layer. The PCA preserves the global structure, while t-SNE emphasizes local relationships in the data. From Figure 9(a), we observe that representations from sensors located at the same body position are clustered closely together, while representations from different body locations are clearly separated. This suggests that our model is signal-agnostic, as it can recognize the signal type differences, map their representations appropriately in the embedding space, and guide feature extraction within each Transformer block.

## F.2    WAVEFORM VISUALIZATION

Figure 9 (b) under "Feature Associations" shows the features extracted by our model. Each patch corresponds to a representation with a vector size of $\mathbb{R}^{768}$. When ordered by time sequence, these representations form 768 waveforms per layer, representing the model's extracted features. The figure displays 64 randomly sampled waveforms from a selected layer. The features highlighted in purple and gray indicate the top 10 patterns positively and negatively associated with the target task (diabetes classification, in this example), with associations determined by linear regression parameters during linear probing. Additionally, our relevance-based fusion mechanism identifies the contribution of each time step during inference, highlighted by red dots in the "Time Step Relevance" section of Figure 9 (b).

Such a visualization pipeline can assist researchers and clinicians by offering insights into how the model reaches its final predictions. Given the millions of parameters and hundreds of waveform features per layer, visualizing these features individually is inefficient for understanding the overall behavior of the proposed foundation model. As a result, we use several techniques in nonlinear dynamic analysis (Thompson et al., 1990) to quantify the overall patterns of these extracted features, which are discussed in detail in section 3.5.

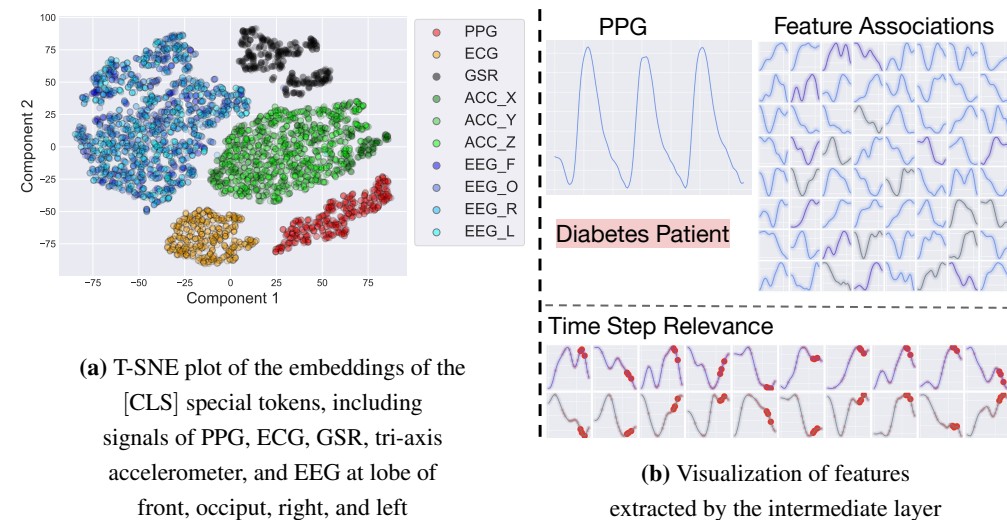

(a) T-SNE plot of the embeddings of the [CLS] special tokens, including signals of PPG, ECG, GSR, tri-axis accelerometer, and EEG at lobe of front, occiput, right, and left

(b) Visualization of features extracted by the intermediate layer

Figure 9: Feature visualization.

## F.3 T-SNE PLOT AMONG CLASSES

In this section, we present T-SNE plots of NormWear's embeddings across different classes to provide insights into their structure and assess their suitability for sample similarity-based information retrieval. It is important to note that these plots are exploratory in nature and do not serve as a claim of the embeddings' superiority. As shown in Figures 10, 11, 12, clear class separations can be observed in certain scenarios. For example, EEG samples from seizure subjects and normal subjects are distinctly separated, and physical activity types are well-clustered. For ECG data, abnormal heartbeats tend to form cohesive clusters. However, it is essential to recognize that these T-SNE plots reduce the latent representations into a 2D space, which may not fully capture the inherent properties of the embeddings in their original high-dimensional form.

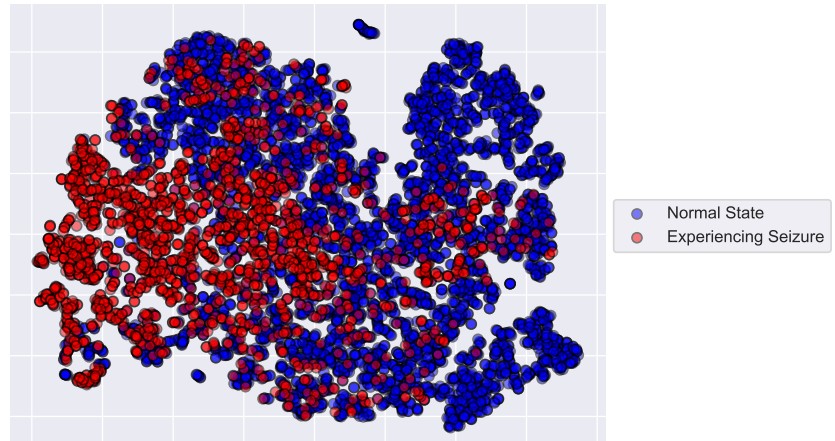

Figure 10: Visualization of embedding on EEG signals.

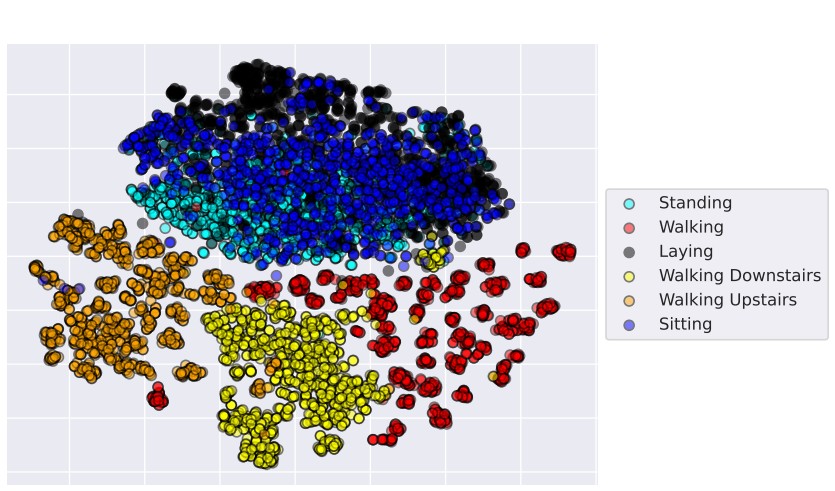

Figure 11: Visualization of embedding on signals from IMU sensors.

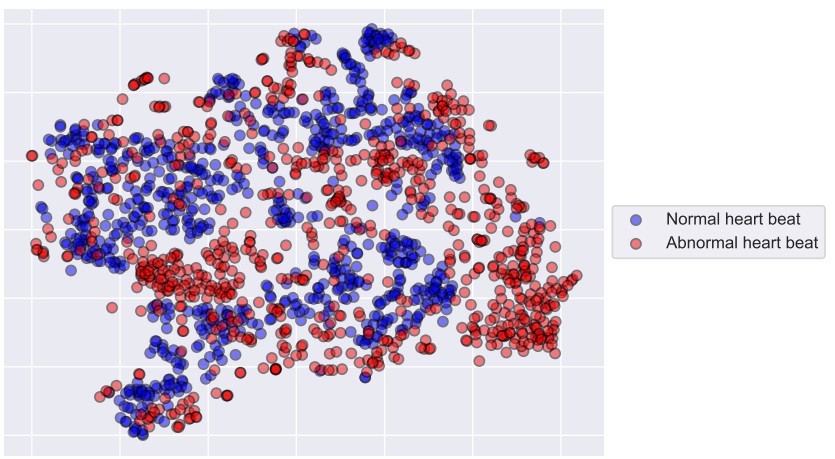

Figure 12: Visualization of embedding of ECG.

# G    RECONSTRUCTION EXAMPLE

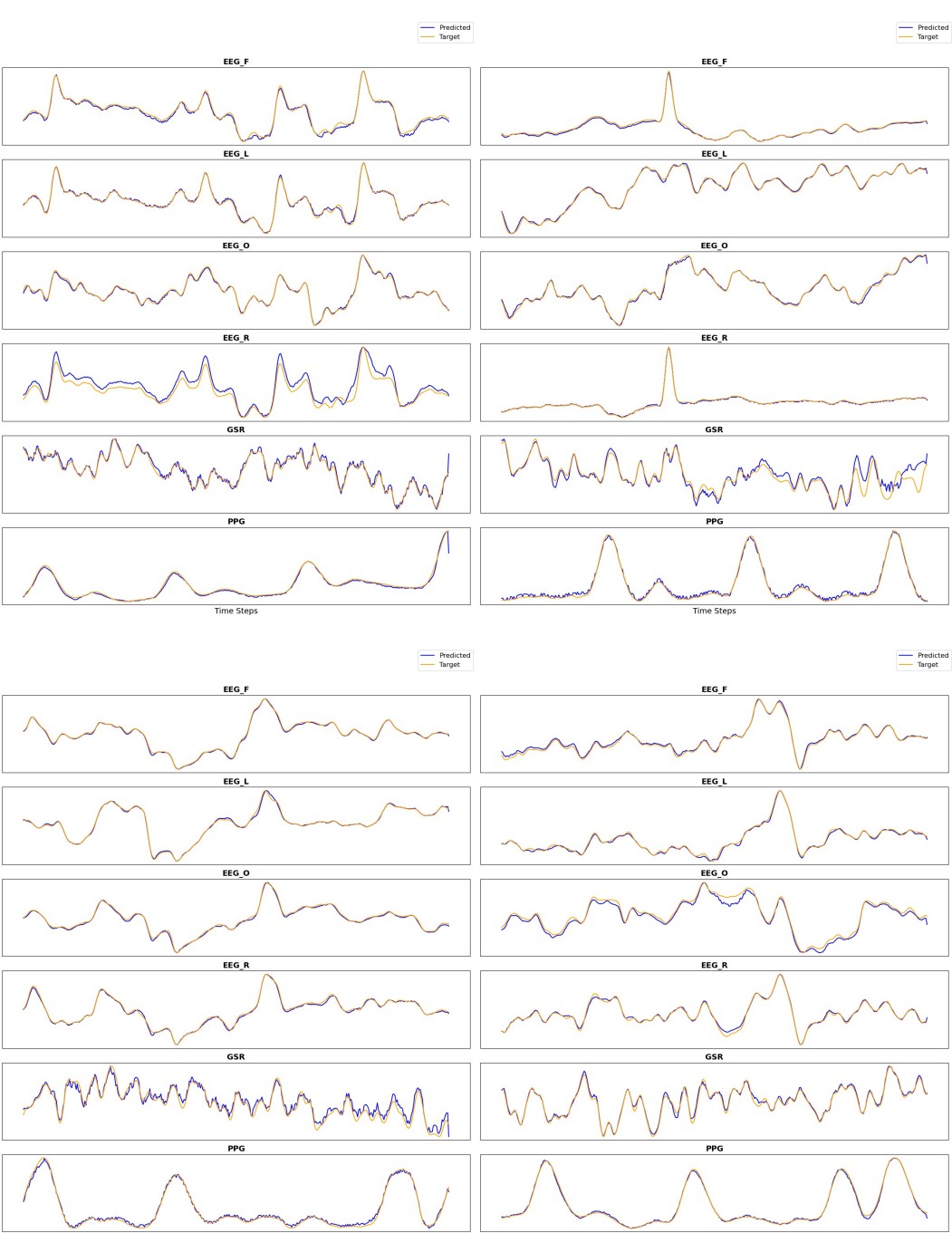

Figure 13: **Uncurated random samples** on Phyatt scalogram, using a NORMWEAR trained in our training set. The masking ratio is 80%.

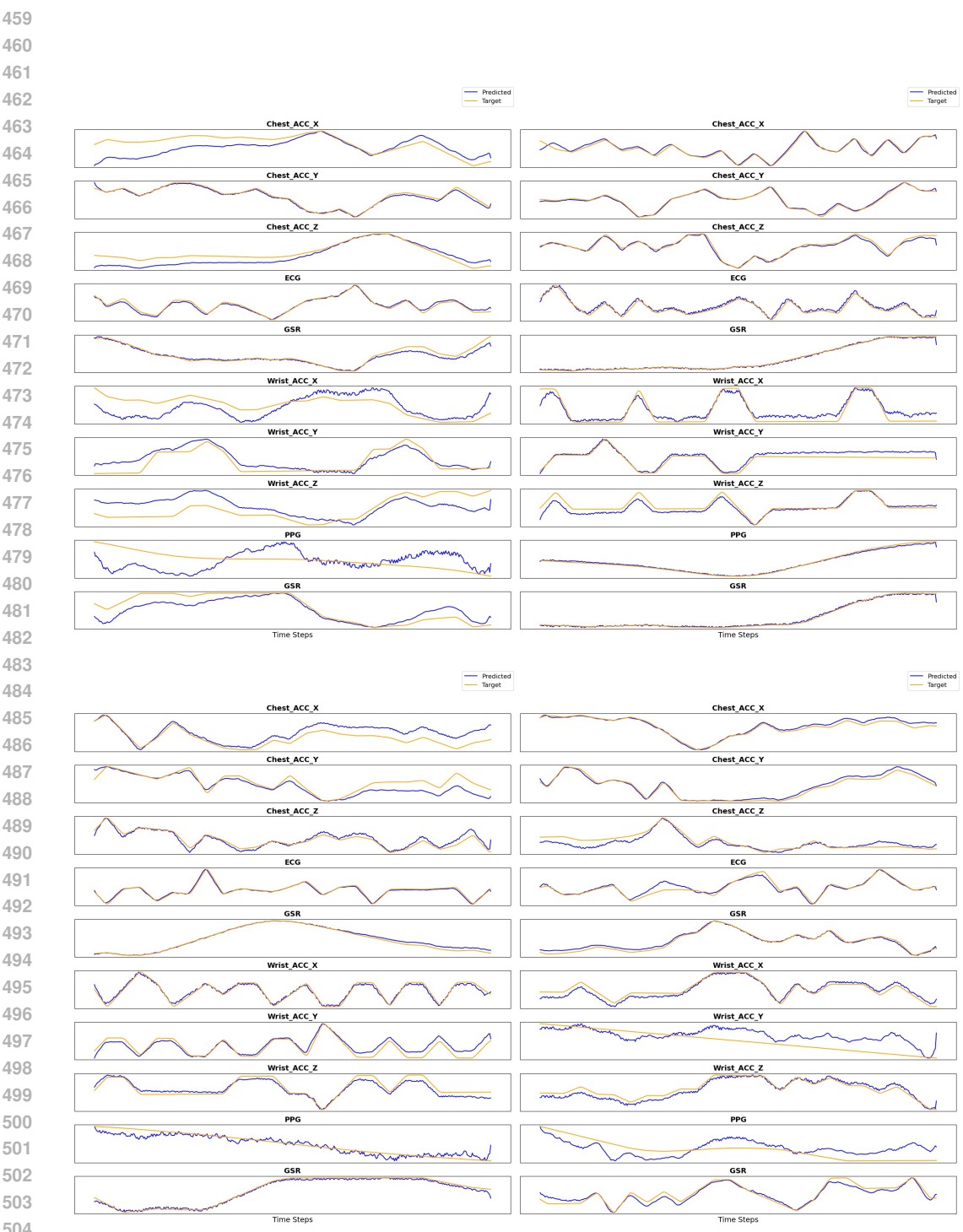

Figure 14: **Uncurated random samples** on WESAD scalogram, using a NORMWEAR trained in our training set. The masking ratio is 80%. Note that the IMU data are not in the training set and, in general, NORMWEAR is able to reconstruct this with high accuracy.

