# OpenReview forum: "Toward Foundation Model for Multivariate Wearable Sensing of Physiological Signals"
_ICLR.cc/2025/Conference — Submitted to ICLR 2025_

### Official Review · Reviewer_hHPV · 2024-10-22

**Soundness:** 2
**Presentation:** 3
**Contribution:** 2
**Rating:** 3
**Confidence:** 4

**Summary:**

This paper presents and evaluates a general methodology for pre-training foundation models (intended for use with physiological signal streams) that can accept multivariate inputs and output model embeddings that are useful for multiple downstream tasks, all related to physiological measurements.   The model family (NormWear) is novel and original in that it is specifically intended to accept multiple signal inputs and behavior in a signal-agonistic manner.

Downstream performance evaluation is reported for 12 different tasks using several different publicly-available data sets.  Additionally, the authors describe a method for using NormWear in a zero-shot learning context and report performance for the 12 downstream tasks.  Lastly, the authors describe two strategies aimed at enabling model interpretability:  analyzing ‘Feature Associations’ and ‘Time Step Relevance’, as well as nonlinear dynamics (chaos system analysis) within the model layers using different signal inputs.

**Strengths:**

This paper represents strong contributions in the following areas:

**Originality:** This idea is novel among publications related to physiological/biological sensing.  I have not seen any past published examples of work (in this domain) aimed at developing foundation models for a variety of multi-modal signal inputs.

**Clarity:**  The authors clearly communicated their methods and objectives for evaluating performance on downstream tasks.


**Additionally the authors should be commended for two important things:**
1. They leveraged public and freely-available data sets and provided enough information/references for a reader to locate the corresponding data, making it straightforward for a reader to obtain the data and do their own analysis on it.
2. They shared their codebase directly in the submission, making it possible for the reader to understand in detail how they did the performance evaluations on each downstream task.

**Weaknesses:**

# Additional Discussion Details #

## Issue 1: Weak Baselines

In Section 3.1 and Table 1, the authors report that NormWear achieves the best performance across tasks.  However, this is limited in part by their choice of baseline models (and general modeling approaches) to compare against NormWear.    This makes it very difficult to determine whether the novel method (NormWear) actually represents a meaningful improvement in downstream task performance vs. representing a small or negligible improvement over a poor reference baseline.


### Potential ideas addressing this issue:
In general, novel modeling approaches should be compared against simple baseline methods such as logistic regression, random forest, or even constant predictor (“guessing the mean”), in addition to comparing against SOTA methods). In regression tasks, the performance for a mean predictor can indicate the “floor” for performance without utilizing any modeling.

## Issue 2: Poor model performance compared to simple baselines

Fortunately, the authors utilized open and freely-accessible data sets for their downstream evaluation. This made it possible for me to spot check the performance of very simple baseline models on several of the tasks.  Due to time limitations I could not do this analysis for every task, but was able to do it for a majority of tasks (7 of the 12 tasks listed in Table 1).  I took care to use the same data set (sourced from the authors’ reference list in Appendix A) and used identical performance metrics reported by the authors (leveraging their accuracy metrics calculations for regression and classification tasks as shown in the evaluate() function on lines 123-138 of engine_linprob.py).  For the following tasks and datasets I observed equivalent or superior performance to what is reported in Table 1, using a very simple method in each case:

Hemoglobin Estimation (regression): I observed accuracy = 88.68 using a constant (mean) predictor. This is significantly better than the best NormWear model, indicating that NormWear performs worse than simply guessing the mean.

Fetal Heart Rate Estimation (regression): I observed accuracy = 96.31 using a constant (mean) predictor.  It is also possible to achieve accuracy=96.38 using a single value (140.0) that I obtained by googling “what is typical fetal heart rate at 20 weeks”.  Both of these are higher accuracy than the best NormWear model.

Blood Pressure Estimation (classification):  I observed accuracy = 91.49 (statistically equivalent to the best NormWear model) using a simple bivariate linear regression model with terms for Age and BMI .

For the following 4 Risk Evaluation tasks I used a simple logistic regression model (sklearn.linear_model.LogisticRegression) with only demographic inputs (Age, BMI, sex):

Hypertension Risk Evaluation (4-class classification): ROC AUC = 0.720, significantly better than the best NormWear model.
Diabetes Risk Evaluation (binary classification): ROC AUC = 0.672 (not as good as NormWear with CLS attention, but significantly better than all other baselines).
Brain Stroke Risk Evaluation (binary classification): ROC AUC = 0.792, significantly better than the best NormWear model.
Brain Disease Risk Evaluation (3-class classification): ROC AUC = 0.779, significantly better than the best NormWear model.


Additionally, in the zero-shot performance (Table 2) for several tasks even the best model does not perform much better than random guessing.  For example in the ‘Heartbeat abnormal Detection’ task all models achieve ROC AUC <0.5 (worse than guessing).  For Emotion Classification, Valence-Arousal Prediction, Driver Fatigue Detection, Hypertension Risk Evaluation and Diabetes Risk Evaluation no models achieve ROC AUC>0.60.  For hemoglobin estimation and fetal heart rate estimation, all zero-shot models perform significantly worse than simply guessing.

This poor level of performance suggests that the models may not actually be learning anything relevant for zero-shot inference.

### Potential ideas to address this issue:
Choose an adequately strong and simple/interpretable baseline model for each task-- for example, this could even be as simple as using a mean predictor on the regression tasks, or using an age-based predictor for the classification tasks.  Then compare the performance of each new model against the simple baseline. Highlight cases that represent significant performance improvements over the simple baseline.  For downstream tasks that show no improvement over the baseline, consider removing these from the paper (or doing additional experiments and development on the model until it significantly outperforms the baseline).


## Issue 3: Lack of discussion relating to visualization/interpretability

In figure 4 the authors present a graphical summary of their visualization and model interpretation analysis.  However, the discussion does not provide any evidence that the interpreted features are useful.  For example, in Figure 4b the Feature Associations and Time Step Relevance (for a Diabetes PPG sample) do not appear to relate to either the input PPG signal or do diabetic physiology.   There is no discussion providing guidance on how to relate the visualized features to the model’s prediction or the target class.

### Potential ideas to address this issue (feature associations and time step relevance):
The authors should provide a more comprehensive analysis of the model interpretability.  For the PPG risk evaluation tasks, this should consist of comparing feature associations for all PPG examples in the PPG data set, split according to the target task.  If the feature associations differ significantly and quantiatively for two target tasks (for example hypertension classification vs. diabetes classification) that would provide some evidence that the model utilizes different PPG features for different objectives.  For the Time Step Relevance, perform some analysis using all PPG examples indicating quantiatively whether the Time Step Relevance consistently highlights known PPG features (for example diastolic foot or systolic rise in the waveform).

In Figure 4a, the T-SNE plot of the embeddings of the [CLS] special tokens for each signal type show clear clustering by type.  However, the authors do not link this in a quantitative way that explains why this clustering makes the model “signal agnostic”.  This clustering according to signal type that is displayed may also be achievable with a short list of signal-level metrics such as signal mean, standard deviation, skew/kurtosis, or power content in several frequency bins.  It would be helpful if the authors could provide some quantitative (or even visual) comparison of clustering using an alternative approach (such as manual feature engineering), in order to demonstrate that the model embeddings are superior.

### Potential ideas to address this issue (T-SNE):
It would be helpful if the authors could provide some quantitative (or even visual) comparison of clustering using an alternative approach (such as manual feature engineering), in order to demonstrate that the model embeddings are superior.



## Issue 4: Small scale of data sets used for pre-training
Adding up the total number of data examples in Appendix A Table 3, I count only ~37,500 examples used for pre-training.  These samples are just 6 seconds long, so the total pre-training data volume is only 62 hours of data (from <1100 subjects).  Given that this data is used to pre-train a model with many millions of parameters, there is a significant risk of overfitting.  The data scale and complexity may not be well suited for the chosen model complexity.

Additionally, for some signal types the total number of unique subjects represented in pre-training is very small— for example all EEG data in pre-training comes from only 45 unique subjects.  This seems likely to introduce some limitations to the pre-training data domain, and increase likelihood of significant domain shift when the model is applied to other small-N independent data sets in downstream tasks.


### Potential ideas to address this issue:
At a minimum, include discussion of the limitations associated with the relatively small data set used for developing a multi-modal foundation model, and the potential impact on generality.  Alternately, provide some quantitative evidence (for example, experiments showing performance as a function of model parameters) indicating that the model complexity is well suited to the available training data.

For downstream tasks that involve data sets containing a small number of subjects (such as Driver Fatigue Detection, N=12) consider utilizing k-fold cross-validation stratified by subject ID to report performance, rather than using a fixed 20% test set.

**Questions:**

I have the following comments for their authors regarding presentation style and content:

## Performance table should have consistent content throughout the document.

References should match with the data source, and be shown in the table.  For example Table 1 lists the reference for the data source in the table (this is preferred), but Table 2 does not.  Appendix A Tables 3 and 4 list the references for the data source only in the table caption, but this should be within the table as done for Table 1.

Include a note (in the performance tables) indicating whether the task is classification vs. regression.  Ideally also include a list of the input data signals that have been used for each task, since this may not always match the full set of signals in that data set (as listed in Tables 3 and 4).

## Several references contain insufficient information
Examples: Liang 2018, Bousseljot 1995, Jianliang Min 2017.  At a minimum, references should include ae DOI or URL. Preferably, cite the primary journal article (if available) in standard citation format.

## Label y-axis plots with numeric values and units (even if units are arbitrary, label these as A.U.)
The reconstruction plots in Appendix Section F should include units on the y-axes. For some examples (e.g. accelerometer) it is clear that the y-scale is likely much different from other examples, but this is impossible to know for sure because the axes are not labeled with units.

## Data preprocessing details are too sparse
Include more information on data preprocessing.  What bandpass filtering parameters were used, if any?   What were the de-trending and gaussian smoothing parameters?  Were these identical for all data sets?

Discuss the limitations of resampling to 65Hz for ECG signals (this loses some meaningful physiological information).

---

> ### Author Response · Authors · 2024-11-21
> **Response to Reviewer hHPV Part I**
>
> Thank you for your helpful feedback! We appreciate that you find our method novel and recognize the clarity of our communication. We are also grateful for your acknowledgment of the transparency in our work, including the use of public datasets and the sharing of our codebase, which we hope will facilitate further exploration and analysis. Below, we briefly respond to each of your concerns. If, after reading our responses, you still feel that any aspects of our work could be improved, we would be grateful for your further suggestions to help enhance the paper. We also really appreciated that the reviewer clearly differentiated the issues into sections, so we can address them precisely, one by one. Our response has five sections.
>
> # Section 1: Response to Issue 1
>
> We sincerely thank the reviewer for their detailed feedback and for conducting their own analyses to benchmark our work. We greatly appreciate the effort invested in this review, as it highlights important areas for clarification and improvement. Below, we address the concerns raised:
>
> ### New Baselines
> We have incorporated statistical baselines, CLAP, and TF-C in our evaluation to provide a more comprehensive comparison. These additional baselines offer clearer context for understanding our model's performance. Results, including comparisons with these baselines, have been updated in the manuscript as well. To further clarify our choices: modeling multivariate wearable signals, which include a wide range of modalities such as heart, brain, and physical motion signals, poses unique challenges, as no universally recognized open-source baseline or state-of-the-art model exists in this domain. To address this, we selected baselines from three representative paradigms:
>
> 1. **Statistical Features**
>    Handcrafted statistical features are a traditional yet widely used method in signal processing literature. We included this baseline as a sanity check to benchmark our model against simpler approaches.
>
> 2. **SSL Time-Series Frameworks**
>    Wearable sensory data is inherently sequential, making time-series modeling strategies relevant. We compared our approach with Chronos, a state-of-the-art time-series forecasting framework, and TF-C, a commonly used self-supervised learning method for time-series data.
>
> 3. **Spectrum-Based Methods**
>    Spectrogram-based modeling has demonstrated state-of-the-art performance in tasks such as music classification and physiological signal analysis. CLAP was selected as a representative baseline for this paradigm.
>
> ### Linear Probing Algorithm
> To ensure better reproducibility, numerical stability, and fairness in performance comparison, we have updated our linear probe implementation. Previously, our results were based on a custom linear layer optimized with the Adam optimizer. To standardize the evaluation pipeline, we now utilize well-established methods for solving the linear probe tasks. Specifically:
> - **Classification tasks** are addressed using Newton's method with conjugate gradient.
> - **Regression tasks** (e.g., vital signs prediction) are solved using Cholesky decomposition with a closed-form solution.
>
> These updates provide a more robust and interpretable framework for evaluating our model's representations across various tasks. The updated results is attached in the next part of response, and we've also updated it in the Experiment section.

---

> ### Author Response · Authors · 2024-11-21
> **Response to Reviewer hHPV Part II**
>
> # Section 1: Response to Issue 1 - Continued
>
> | **Downstream Tasks**                              | **Statistical** | **Chronos** | **CLAP** | **TF-C** | **NormWear (Ours)** |
> |---------------------------------------------------|-----------------|-------------|----------|----------|---------------------|
> | WESAD                                             | 66.213          | 71.489      | 72.383   | 69.865   | **76.060**          |
> | UCI-HAR                                           | 95.784          | 91.593      | 96.420   | 96.892   | **98.954**          |
> | DriverFatigue                                     | 63.249          | **76.722**  | 61.889   | 66.882   | 74.292              |
> | **Activity Recognition Avg.**                     | 75.082          | 79.935      | 76.897   | 77.880   | **83.102**          |
> |---------------------------------------------------|-----------------|-------------|----------|----------|---------------------|
> | Epilepsy (eye open)                               | 82.489          | 82.41       | 85.094   | 89.153   | **92.743**          |
> | Epilepsy (eye close)                              | 87.457          | 88.218      | 89.867   | 94.416   | 94.828              |
> | Epilepsy (health area)                            | 86.274          | 81.08       | 83.711   | 85.619   | **88.541**          |
> | Epilepsy (tumor area)                             | 82.816          | 81.034      | 83.644   | 86.348   | 87.197              |
> | Epilepsy (seizure)                                | 88.272          | 97.572      | **97.734**| 93.998   | 97.053              |
> | GAMEEMO                                           | 51.009          | 53.747      | 52.551   | **56.275**| 54.937              |
> | **EEG Main Tasks Avg.**                           | 79.720          | 80.677      | 82.100   | 84.302   | **85.883**          |
> |---------------------------------------------------|-----------------|-------------|----------|----------|---------------------|
> | PhysioNet Challenge                               | 42.592          | 48.166      | 43.058   | **50.934**| 46.904              |
> | ECG-Abnormal                                      | 97.092          | 98.585      | 97.23    | 98.275   | **99.140**          |
> | PPG-BP (HTN)                                      | 59.499          | 52.425      | 56.757   | **65.229**| 62.341              |
> | PPG-BP (DM)                                       | 47.823          | 51.164      | 42.455   | **57.883**| 55.893              |
> | PPG-BP (CVA)                                      | **71.25**       | 50.278      | 51.667   | 58.125   | 70.625              |
> | PPG-BP (CVD)                                      | 51.219          | 58.31       | 50.91    | **58.674**| 51.773              |
> | PhysioNet EMG                                     | **99.309**      | 61.6        | 98.627   | 78.308   | 99.216              |
> | **Risk Evaluation Avg.**                          | 66.969          | 60.075      | 62.958   | 66.775   | **69.413**          |
> |---------------------------------------------------|-----------------|-------------|----------|----------|---------------------|
> | Noninvasive-BP                                    | 92.31           | 91.79       | 91.922   | 87.481   | **92.420**          |
> | PPG-Hgb                                           | 94.219          | **95.005**  | 94.291   | 93.408   | 94.632              |
> | Fetal-fPCG                                        | 98.929          | 99.048      | **99.195**| 99.077   | 99.072              |
> | **Vital Signs Avg.**                              | 95.153          | 95.281      | 95.136   | 93.322   | **95.375**          |
> |---------------------------------------------------|-----------------|-------------|----------|----------|---------------------|
> | **Micro Avg.**                                    | 76.727          | 75.276      | 76.284   | 78.255   | **80.875**          |
> | **Macro Avg.**                                    | 79.231          | 78.992      | 79.273   | 80.570   | **83.443**          |

---

> ### Author Response · Authors · 2024-11-21
> **Response to Reviewer hHPV Part III**
>
> # Section 2: Response to Issue 2
>
> We truly appreciate the reviewer’s time and effort in evaluating our model, especially for actively testing our dataset, which speaks to their commitment to a thorough review. We understand the challenges inherent in small datasets, and while we are grateful for the feedback, we do feel that drawing strong conclusions based solely on these limited examples may not offer a complete picture of our model’s capabilities.
>
> On the topic of demographic features like age and BMI, we respectfully acknowledge that these variables are important for application-specific modeling. However, as our work focuses on advancing representation learning for wearable sensor data—without relying on demographic inputs—this falls outside the scope of our study. As pioneers in developing a foundation model for this type of data, our primary goal is to establish a versatile, scalable approach. Application-specific improvements are certainly worthwhile, but they are more appropriate for future work that dives deeper into individual tasks. It is important to note that demographic and wearable data capture distinct types of information, and their combination is often addressed in more specialized research, which, as we noted, is not the focus of our study.
>
> We acknowledge that our zero-shot performance is not yet promising at this stage, as discussed in the limitation section. However, we hope this initial attempt provides valuable insights for the community and practitioners on how to leverage the representations from our signal encoder. It is clear that it will not perform on par with state-of-the-art signal-text alignment models, as those models are trained from scratch with paired input, which is difficult to obtain in wearable sensory data. Therefore, the lightweight connector module we proposed is a preliminary attempt, given these practical constraints. We look forward to further research and improvements in this direction and hope that our work can inspire future advancements.
>
> # Section 3: Response to Issue 3
>
> This is not our primary focus. The use of `[CLS]` is simply a way to highlight that the model is agnostic to the type of sensor input. The feature relevance and waveform visualizations are meant to set up the subsequent analysis with nonlinear dynamic metrics (NLD), which provides insights into how the model behaves across different layers. This approach allows us to quantify and examine the features learned by the model—for example, shallow layers capturing more periodic features, and deeper layers capturing more unpredictable patterns. We are not claiming superiority in terms of embeddings but using this method to analyze the model’s behavior.
>
> We acknowledge that the figures could have been misleading, and we have relocated them to the appendix to clarify their purpose. This ensures that the main text now focuses on key aspects like model performance, technical details, and a discussion of limitations.

---

> > ### Comment · Reviewer_hHPV · 2024-11-25
> >
> > I thank the authors for the extensive additional experiments that they performed in support of the latest manuscript version, as well as for their clear explanation of the new material.  **However I still have a significant number of major concerns with the work, listed below:**
> >
> > ## Baselines and Demographics
> >
> > My use of demographic inputs in low-complexity models applied to the publicly-available data sets was mainly to determine whether the targets in question could be predicted from demographic information more accurately than the models in the authors’ table.  ** Demographics (age, sex, BMI, race/ethnicity) can be estimated with high accuracy by most of the signal modalities covered in the manuscript** (detailed analysis of demographics prediction accuracy by ECG and PPG is reported in [1]; analysis of demographics prediction using IMU signals is reported in [2]; many other past research publications have reported age and sex prediction accuracy using IMU, EEG and audio signals).
> >
> > The importance of reporting accuracy on downstream demographic targets is to provide a quantitative indication of how well each pre-trained encoder can extract features from each sensing modality that capture demographic information.  The importance of comparing sensor-derived prediction accuracy on various downstream targets with demographics-derived prediction accuracy on the same targets, is in order to provide an indication of whether the predictive capability of the authors’ models on these downstream tasks extends beyond just relying on demographics features as prediction ‘shortcuts’.   **Downstream performance on several of the tasks in Table 3 are within the range where the performance may be explained by a model’s reliance on features capturing demographic information.** Specific examples falling into this category that I have checked by analyzing the available data sets are Noninvasive BP regression and the PPG-BP risk estimation tasks.
> >
> > ## Unexpected (negative) performance changes on some tasks after adding more pre-training data
> >
> > The authors report that a significant difference in their latest manuscript version, compared with the original, is the use of over 6 times more data in pre-training (or over 60 times more, if counting augmentation).  This increase in pre-training data volume has translated into higher performance on average for several downstream tasks.  However, I notice that for some of the PPG-BP classification tasks (specifically HTN and DM tasks) **the performance is now significantly worse than reported in the original paper.**  V1 NormWear **performance was 66.9 and 72.2 on HTN and DM classification**, respectively, using linear probing.  The latest NormWear performance **has now dropped to 62.3 and 55.9 on these two tasks**, respectively.  **This significant drop in downstream task performance despite the addition of a large volume of pre-training data is counterintuitive, and calls into question the generality and robustness of the authors’ overall approach.**
> >
> >
> > ## Concerningly high level of variability in performance on similar downstream tasks (for example heart rhythm inference from ECG)
> >
> > The single-lead-ECG AF classification performance on the PhysioNet challenge data is equivalent to (or even worse than) random guessing on all models listed in the table, with ROC AUC values in the range of 40-50.  In contrast, the conceptually similar task of classifying abnormal heart rhythm from multi-lead ECG (ECG-Abnormal task) produces high performance numbers (ROC AUC 97-99) across all models including the statistical baseline.  Both tasks consist of binary classification using the same input modality (ECG). **The extremely variable performance on these two similar tasks, ranging from “worse than guessing” to “excellent”, raises concerns regarding the generality and robustness of the overall approach.**
> >
> >
> > ## Zero-shot performance concerns:
> >
> > The material on zero-shot prediction, although it contains novel ideas that should definitely be explored in future work, also seems to have some major limitations.  Most notably, it does not seem to make any useful predictions for almost all of the tasks.  With the exception of UCI-HAR and Noninvasive BP, the zero-shot performance on classification tasks is roughly equivalent to random guessing (ROC AUC’s in the 40’s and low 50’s) and on regression tasks is not better than a constant predictor. As mentioned in my initial set of comments, for the PPG-Hgb task I was able to get MRE of 88.68 using a constant (mean) predictor, and for the fPCG regression task I was able to get MRE of 96.4 using a constant predictor (140 bpm, a common fetal HR).  These values are better than the zeros-shot performance reported from any method in the table. **Despite being an admirable goal and an area worthy of continuing effort, overall the material on zero-shot inference does not seem mature enough to include in a research manuscript (mainly because it doesn’t work yet).**

---

> > > ### Comment · Reviewer_hHPV · 2024-11-25
> > > **References corresponding to above comment**
> > >
> > > ### References
> > > [1] Abbaspourazad, S., et al. (2023). Large-scale training of foundation models for wearable biosignals. arXiv preprint arXiv:2312.05409.
> > >
> > > [2] Spathis, Dimitris, et al. "Self-supervised transfer learning of physiological representations from free-living wearable data." Proceedings of the Conference on Health, Inference, and Learning. 2021.

---

> > > > ### Comment · Reviewer_hHPV · 2024-11-25
> > > > **Summary**
> > > >
> > > > ### Summary
> > > > The above issues highlight a general concern with the methodology and the maturity of the analysis.  Although the general objectives of this work, as well as the proposed direction for future work as summarized in the ‘Limitations and Conclusion’ section) give me optimism that it may translate into high-quality impactful contributions in the future, in its current state I do not feel that this work is mature enough for acceptance.

---

> > > > > ### Comment · Reviewer_hHPV · 2024-11-25
> > > > > **Added note**
> > > > >
> > > > > Lastly, one remaining minor issue that I feel is important for the authors to be aware of:
> > > > >
> > > > > The authors state in several places that they would like to extend the modeling framework to include audio signals.   In one place they state *“We also suggest extending the proposed model … such as audio … which could provide more thorough health-related information.”*   I would like to highlight for the authors that the fetal phonocardiography signals from Bhaskaran, et al. are in fact audio signals (albeit collected using a contact microphone).

---

> > > > > > ### Author Response · Authors · 2024-11-28
> > > > > > **Response to Reviewer hHPV**
> > > > > >
> > > > > > Thank you for your timely reply and insightful comments. We really appreciated that the reviewer acknolwedged our improvement on this work. In order to address the reviewer's concerns, we've managed the following response and clarification with new experimental results, and we hope this will resolve the main concerns.
> > > > > >
> > > > > > ## Firstly, it is not a drop of performance, it is a result from an updated linear probing pipeline.
> > > > > > We would like to clarify that the observed changes in performance stem from an update in our evaluation pipeline. We change the linear optimizer from Adam with to a more stable and reproducible framework based on Newton's method for classification tasks and Cholesky decomposition for regression tasks as stated in details in our previous response as well as in the updated manuscript. This change was made to ensure numerical stability, fairness, and transparency across evaluations. While it is expected that different evaluation methodologies may yield varying results, we believe the updated framework provides a clearer and more reliable comparison of our model’s capabilities. Importantly, our model continues to demonstrate strong performance overall, even under these stricter evaluation conditions.
> > > > > >
> > > > > > ## Secondly, the high variability in ECG abnormal detection task is due to the issue of one of the dataset
> > > > > > Thank you for raising this concern. We would like to clarify that the two tasks, while both involving ECG data, differ significantly in their nature and complexity. The ECG-Abnormal task focuses on distinguishing normal heartbeats from abnormal ones, mainly myocardial infarction. In contrast, the AF classification task targets atrial fibrillation and involves four classes, one of which is notoriously noisy and difficult to classify, as noted in prior work [1] (also the performance on this data is not reported in [1] which is the source of data that we use, thus, the issue lies in target variables remain unknown). This data quality issue likely explains the universally poor performance (ROC AUC ~0.5) across all models, rather than reflecting a limitation of our approach. Given that our focus is on developing a generalized pretraining model applicable across a wide range of tasks, we have decided not to optimize this specific dataset further within the current scope. To avoid potential misunderstandings, we have decided to remove this task from the manuscript, and all the results table are updated accordingly.
> > > > > >
> > > > > > ### Reference
> > > > > > [1] Zhang, Xiang, et al. "Self-supervised contrastive pre-training for time series via time-frequency consistency." Advances in Neural Information Processing Systems 35 (2022): 3988-4003.

---

> > > > > > > ### Author Response · Authors · 2024-11-28
> > > > > > > **Continue Response to Reviewer hHPV**
> > > > > > >
> > > > > > > ## Thirdly, regarding the Baselines and Demographics
> > > > > > > We thank the reviewer for the clarification on validating the methodology with demographic features, and we now understand the main motivation behind it. We've run the experiment on our downstream datasets that have demographic data available, and demostrate that our model extract complementary information with demographics information, which further proved that wearable signal data and demographics are essentially two separate modalities, and the fusion of modeling these two modalities are another big topic that is out of scope of the focus of this work.
> > > > > > >
> > > > > > > | **Downstream Tasks**     | **Simple Baseline Mode and Mean** | **Demographic** | **NormWear-Medium** | **Demographic + NormWear-Medium** | **NormWear-Large** | **Demographic + NormWear-Large** |
> > > > > > > |---------------------------|----------------------------------|-----------------|---------------------|----------------------------------|--------------------|----------------------------------|
> > > > > > > | **WESAD**                 | 50.000                           | 49.907          | **74.227**          | 69.06                            | **76.06**          | 68.755                           |
> > > > > > > | **Noninvasive**           | 92.988                           | 92.954          | 91.427              | 90.84                            | 92.42              | 92.528                           |
> > > > > > > | **PPG-Hgb**               | 94.816                           | 95.634          | 94.911              | 95.835                           | 94.632             | 96.384                           |
> > > > > > > | **Fetal-fPCG**            | 99.033                           | 99.039          | 98.997              | 99.001                           | 99.072             | 99.097                           |
> > > > > > > | **Vital Signs Avg.**      | 95.612                           | **95.876**      | 95.112              | 95.225                           | 95.375             | **96.003**                       |
> > > > > > > | **PPG-BP (HTN)**          | 50.000                           | 59.899          | 62.746              | 64.482                           | 62.341             | 61.291                           |
> > > > > > > | **PPG-BP (DM)**           | 50.000                           | 47.297          | 62.613              | 47.86                            | 55.893             | 60.135                           |
> > > > > > > | **PPG-BP (CVA)**          | 50.000                           | 81.875          | 67.639              | 83.681                           | 70.625             | 77.847                           |
> > > > > > > | **PPG-BP (CVD)**          | 50.000                           | 71.011          | 51.504              | 70.37                            | 51.773             | 67.466                           |
> > > > > > > | **Risk Evaluation Avg.**  | 50.000                           | 65.021          | 61.126              | **66.598**                       | 60.158             | **66.685**                       |
> > > > > > > | **Micro Avg.**            | 67.105                           | 74.702          | 75.508              | **77.641**                       | 75.352             | **77.938**                       |
> > > > > > > | **Macro Avg.**            | 65.204                           | 70.268          | 76.821              | **76.961**                       | **77.198**         | 77.148                           |
> > > > > > >
> > > > > > >
> > > > > > > This result table is also updated in the manuscript under Appendix D, Table 12. Is shows that demographic information and represenations extracted from wearable signals have their own strength on different tasks, and most of the time, when we concat them together, the overall performance will be better, just as we stated in previous response that they are often used together when dive deeper on each specific application. The performance drop in some cases after concatenation, which indicate that there might be some confounding relationship between these two representations, hence further indicated that the information lies in demographic and the wearable represenation from NormWear are focused on different aspects. Same observation are observed with arbitrary model checkpoints during pretraining (denoted as Medium and Large marker representing different stage of training when we do the study on increasing the pretrain size.)

---

> > > > > > > > ### Author Response · Authors · 2024-11-28
> > > > > > > > **Continue Response to Reviewer hHPV**
> > > > > > > >
> > > > > > > > ## Finally, regarding the zero shot performance
> > > > > > > > We totally understand the concern on the zero-shot performance near 0.5 of AUC ROC, and we would like to share our most recent results on zero shot performance after we train on the larger pretrain size with more text augmentation. We also include the aligned performance from previous checkpoints reported in previous version of manuscript this time as "w/o text augmentation" for a better view of the improvement. The following table is also updated in the main section under section 3.4:
> > > > > > > >
> > > > > > > > | **Model**                     | **WESAD** | **UCI-HAR** | **DriverFatigue** | **GAMEEMO** | **Epilepsy (eye open)** | **Epilepsy (eye close)** | **Epilepsy (health area)** | **Epilepsy (tumor area)** | **Epilepsy (seizure)** | **PPG-BP (HTN)** | **PPG-BP (DM)** | **PPG-BP (CVA)** | **PPG-BP (CVD)** | **ECG-Abnormal** | **PhysioNet EMG** | **Micro Avg.** | **Macro Avg.** |
> > > > > > > > |-------------------------------|-----------|-------------|-------------------|-------------|------------------------|--------------------------|----------------------------|---------------------------|------------------------|------------------|-----------------|-----------------|-----------------|-----------------|-----------------|----------------|----------------|
> > > > > > > > | **CLAP - MD**                  | 45.3      | 62.8        | 58.5              | **53.1**    | 44.9                   | 45.1                     | 47.6                       | 30.5                      | **84.9**               | **59.4**          | 41.8            | 46.0            | 57.4            | 22.9            | 55.4            | 50.4           | 51.2           |
> > > > > > > > | **CLAP - DP**                  | 50.7      | 52.3        | **61.1**          | 51.6        | **54.4**               | 41.9                     | 58.6                       | 46.4                      | 74.3                   | 52.2             | 41.4            | 50.6            | 58.9            | 42.7            | 38.3            | 51.7           | 52.2           |
> > > > > > > > | **NormWear w/ MSiTF**          | 55.9      | **71.4**    | 54.9              | 50.2        | 54.0                   | 56.4                     | **66.9**                   | **57.4**                 | 53.7                   | 56.5             | 53.2            | **65.0**         | **63.1**         | **74.3**         | **65.7**         | **59.9**        | **60.1**        |
> > > > > > > > | **NormWear - w/o IMP**         | **56.2**  | 70.3        | 55.4              | 49.8        | 54.0                   | **56.5**                 | **66.9**                   | 57.3                      | 52.9                   | 56.5             | **54.3**         | 61.7            | 60.7            | 73.4            | 65.2            | **59.4**        | **59.6**        |
> > > > > > > > | **NormWear - w/o text aug**     | 54.8      | 65.8        | 55.2              | 49.2        | 31.0                   | 58.4                     | 58.6                       | 32.8                      | 58.1                   | 50.2             | 52.6            | 50.8            | 50.6            | 47.7            | 33.6            | 50.0           | 51.4           |
> > > > > > > >
> > > > > > > > For the 3 regression tasks on vital sign estimation, we exclude them from the zero-shot for now, because we've notice that our zero-shot modeling approach is designed to better fit for classification scenario. As a response to this aspect, we've add a discussion in the limitation section to indicate this potential future research direction.
> > > > > > > >
> > > > > > > > We hope our additional results and clarification could better address the reviewer's concern, and please let us know if there are any confusions. Thank you again for you constructive comments!

---

> ### Author Response · Authors · 2024-11-21
> **Response to Reviewer hHPV Part IV**
>
> # Section 4: Response to Issue 4
>
> We appreciate the reviewer’s insightful comment regarding the size of the pre-training dataset. To address this concern, we have significantly expanded both the pretraining and downstream evaluation datasets:
>
> ### Pretraining Data
> Originally, for computational efficiency, we had not utilized all available samples in our pre-training dataset. To address this, we have now included all available samples and supplemented the dataset with several additional collections, including datasets containing both ECG and PPG signals, as well as a large multi-channel EEG dataset with over 50,000 samples. As a result:
> - The pre-training dataset has grown from **37k samples (62 hours)** to **241k samples (402 hours)**.
> - Leveraging time series augmentation techniques detailed in the appendix, we further expanded the dataset to **2.5 million samples (approximately 4000 hours)**.
>
> All experiments were re-run with the model retrained on this significantly larger pre-training dataset.
>
> ### Downstream Data
> We incorporated several new datasets to further validate the model's performance. These include:
> - An EEG dataset with 500 subjects and five tasks.
> - A dataset focused on muscular disorders.
> - A new ECG abnormality detection dataset.
>
> This expanded evaluation framework ensures a more comprehensive assessment of the model across diverse health-related tasks.
>
> ### k-Fold Cross-Validation on Datasets with small number of subjects.
> In response to the reviewer’s suggestion of using k-fold cross-validation stratified by subject ID for tasks involving small number of subjects (such as Driver Fatigue Detection with N=12), we have conducted additional experiments. For these experiments, we selected all downstream tasks with a subject count of 30 or fewer. Importantly, we note that a small number of subjects does not necessarily imply a small number of samples, as some datasets (e.g., Wesad and UCI HAR) have a large number of samples despite fewer subjects.
>
> The results from these experiments consistently show that **NormWear outperforms all baselines**, and these findings are reflected in **Table 12 of Appendix C** in the updated manuscript.
>
> | Downstream Tasks | Statistical        | Chronos            | CLAP              | TF-C              | NormWear-L (Ours) |
> |-------------------|--------------------|--------------------|-------------------|-------------------|-------------------|
> | WESAD            | 79.992 ± 0.707    | 83.332 ± 0.841    | 87.824 ± 0.463    | 82.701 ± 0.536    | **89.585 ± 0.683** |
> | UCI-HAR          | 95.602 ± 0.148    | 91.956 ± 0.256    | 96.864 ± 0.175    | 97.382 ± 0.138    | **98.179 ± 0.06**  |
> | DriverFatigue    | 69.614 ± 1.138    | **72.48 ± 2.848** | 66.251 ± 0.471    | 65.026 ± 1.198    | 68.971 ± 1.32      |
> | GAMEEMO          | 64.281 ± 1.292    | 56.694 ± 0.878    | 64.119 ± 0.543    | 62.925 ± 0.999    | **67.863 ± 0.72**  |
> | Noninvasive      | 92.83 ± 0.386     | 92.223 ± 0.356    | 92.612 ± 0.272    | 88.707 ± 0.622    | **93.381 ± 0.516** |
> | **Avg.**         | 80.464 ± 0.734| 79.337 ± 1.036    | 81.534 ± 0.385    | 79.348 ± 0.699    | **83.596 ± 0.660** |
>
>
> These additions and experiments provide strong empirical evidence supporting the robustness and scalability of NormWear, even when applied to smaller datasets or tasks with limited subjects. We have updated the relevant tables in the manuscript to reflect these expanded datasets and results, ensuring a more comprehensive evaluation.
>
> # Section 5: Response to the Questions
>
> **Q: Discuss the limitations of resampling to 65Hz for ECG signals (this loses some meaningful physiological information)**
>
> A: We appreciate the reviewer’s comment on the resampling of ECG signals to 65Hz. The decision to resample at this rate was primarily motivated by engineering considerations, as it strikes a balance between computational efficiency and maintaining sufficient information for the intended tasks (e.g., heart rate and heart rate variability). For most standard clinical and physiological analyses, such as detecting heart rate or HRV, 65Hz is generally adequate, as these signals do not require the extremely high frequency resolution provided by higher sampling rates.
>
> While higher sampling rates may capture more granular details in the ECG waveform, such as the morphology of certain smaller features, our focus in this work is on broad physiological trends that can be effectively detected at this resolution. We do acknowledge that in future work, when higher resolution or more detailed waveform analysis becomes relevant, resampling at a higher frequency will be explored.
>
> Thank you again for your constructive feedback!

---

### Official Review · Reviewer_9HJ7 · 2024-10-22

**Soundness:** 2
**Presentation:** 1
**Contribution:** 2
**Rating:** 3
**Confidence:** 5

**Summary:**

This paper introduces NormWear, a common foundation model for various physiological signals such as PPG, ECG, EEG, GSR and IMU. Authors have pre-trained NormWear on a dataset containing these different physiological modalities, and performed several downstream comparisons with off-the-shelf time-series and image encoders. They have also investigated different components of NormWear, different modality fusion techniques, and a way to align NormWear embeddings with text embeddings for zero-shot classification.

**Strengths:**

* The idea proposed in this paper is interesting: to have one common foundation model for various physiological signals.
* Authors investigated different components of NormWear in terms of dynamics in different layers, visualizations of features and embeddings.
* Authors investigated different fusion techniques to fuse multi-modal tokens/embeddings.

**Weaknesses:**

* In general, I believe the quality of the writing, presentation and conclusions in the paper can improve significantly. There are several unbacked claims and missing details throughout the paper (see below), which make the paper very hard to follow. I highly suggest authors consider revising the manuscript write up to provide a better flow and additional information. I have done my best to provide several examples in below, but I’m sure there are more improvements that can be made.

* The number of subjects in the pre-training and evaluation datasets makes the conclusions intransferrable to large datasets for claims of “NormWear as a foundation model”. A foundation model is really a generalist model that can perform well on a variety of corner cases and downstream applications. Some modalities (e.g. EEG) have less than 50 pre-training/evaluation subjects, for example, their evaluation of “Driver Fatigue detection” has only 12 * 20% = [2-3] subjects in the test set, which is very low to conclude generalizable performance and conclusions for health applications. I believe this weakens the conclusion of NormWear being “[the first] *foundation model* specifically designed for wearable sensing data, capable of processing any number of multivariate signals from sources such as the heart, skin, brain, and physical body.”. I recommend the author revise the language or provide additional empirical back up for NormWear being a foundation model.
* There are a variety of inadequate references and claims throughout the paper. I recommend authors take a pass through the claims in the paper and revisit them as needed. I provide some examples below:
    * “Despite the great potential of these works across various tasks such as forecasting, anomaly detection, and classification, they are not easily transferable to wearable health applications for two main reasons“: Transformers with images or spectrograms, have been previously used for physiological signals, so authors may reconsider this claim [1], [2].
    * “When modeling this type of data, relying solely on modality-specific backbone feature encoders, such as RNNs (Yu et al., 2019) or transformer-based (Vaswani et al., 2023) neural networks, is insufficient. Therefore, it becomes essential to incorporate established signal processing techniques, such as the short-term Fourier transform (Brigham, 1988) and wavelet transform (Torrence & Compo, 1998)”. It would be great if authors justify these claims. To the best of my knowledge, Transformers (without Fourier transforms) are widely used for physiological signals, and it is not clear to me how transforming the time-series to frequency domain, can remove modality-to-modality variations. If authors provide theoretical/empirical justification for this, it can improve the motivation.
    * “Nevertheless, this method completely ignores information in the frequency domain, leading to significant information loss and suboptimal performance in downstream tasks.“: In my opinion, this is incorrect. Just because a model is trained on time domain, does not mean it *completely ignores information in frequency domain* as there’s a duality between frequency and time domain. I suspect authors may have meant to claim that it’s easier to capture certain frequency-related information if the input in frequency domain is directly given to the model. If yes, it’s a different claim, but please note that a powerful enough encoder with enough data, should be able to capture frequency-related information from time-domain input as well. I recommend authors provide more empirical/theoretical evidence for this claim, or reconsider the writing.
    * “Another important point to consider is that although empirical studies (Nie et al., 2023; Abbaspourazad et al., 2023) show that channel-independent structures effectively capture local patterns, they fail to account for relationships across channels.”: Please provide reasoning for such claims, it’s not clear to me how these conclusions are made from these prior papers.
    * “In order to stay consistent with the literature on foundational representation learner (Devlin et al.,2019; Dosovitskiy et al., 2020; Gong et al., 2021), the backbone of our proposed model consists of a convolutional patching layer followed by 12 standard Transformer blocks (Vaswani et al., 2023).”, there are a lot of different representation learning approaches (masked auto encoder, variational auto encoders, contrastive learning, autoregressive pre-training, ...), so perhaps authors can more accurately rewrite this sentence.
    * “With the state-of-the-art (SoTA) back- bone model for modeling time series data, each intermediate layer will output tensors that contain the timestamp dimension”, what does this mean? Can authors provide back up for this claim or provide more information?
    * “Such a visualization pipeline can assist researchers and clinicians by offering insights into how the model reaches its final predictions” It’s not clear to me whether these visualizations provide any gradient signal or they’re random. To the best of my knowledge, the relationship between PPG and diabetes is not well-understood, so not sure if I can directly conclude that the shown results match with the well-known concepts in the literature. It would be great if the authors can relate this to the literature and present the efficacy of their visualization method.
    * “However, recent works have shown that features extracted from deep learning methods generally outperform handcrafted features in most cases (Yan et al., 2023a; Krizhevsky et al., 2012; Luo et al., 2024).”. I’m not sure how AlexNet is relevant to tokenization discussion in Section 2.2 here, also not very recent :). Can the authors reconsider the discussion here.


* Many important details of technical implementation is missing from the paper, I recommend the authors incorporate all necessary information to aid the reader. I provide few examples below:
    * Information about how patches are selected and how many patches are there for each segment, appear to be missing.
    * Architectural hyperparameters regarding the tokenizer, the reconstruction module (de-tokenization), the details of the encoder/decoder transformer (token dimension, number of attention heads, positional encoding, dimension of MLP hidden layer, normalization, ...) appear to be missing.
    * The details regarding the downstream evaluations (linear probing) appear to missing
    * The details about how sentences are chosen in Section 3.2, what language model (or encoder) was used to get the “question semantic” embeddings appear to be missing.
    * Hyperparameters of equation 1/2 and L295-311 appear to be missing from the paper.
    * Details of masking strategies in Table 8 are missing.
* Several major claims in the paper seem overstated. For example, the delta between NormWear and Chronos in Table 1 seems very small considering that Chronos is not even a proper foundation model on physiological signals (Chronos is just a model trained on some time-series datasets, and to the best of my knowledge, there’s no prior work showing that Chronos is even close to SOTA for physiological signals such as PPG/ECG/EEG). Despite this shortcoming for Chronos, its difference between NormWear in the first 8 evaluations is very small, and in some cases it is even better. Similarly, authors make several big claims about processing frequency domain and CWT (see examples above), however, in Table 9, they show that the difference between processing with CWT vs. raw input is not that much (76.25 vs. 78.27). I recommend authors provide further explanation/discussion regarding these claims.
* It would be great if the authors provide details about how confidence bounds are selected in Tables, e.g., Table 1. It is surprising that they get such narrow confidence bounds with such small N (e.g., 2/3 for Driver Fatigue detection if I understand correctly)?
* Please consider fixing typo and formatting issues, for example:
    * L42: missing space
    * L157: missing space.
    * Table captions not being above the tables.

[1] Mathew, G., Barbosa, D., Prince, J., & Venkatraman, S. (2024). Foundation models for cardiovascular disease detection via biosignals from digital stethoscopes. npj Cardiovascular Health, 1(1), 25.

[2] Vaid, A., Jiang, J., Sawant, A., Lerakis, S., Argulian, E., Ahuja, Y., ... & Nadkarni, G. N. (2023). A foundational vision transformer improves diagnostic performance for electrocardiograms. NPJ Digital Medicine, 6(1), 108.

**Questions:**

* What is the justification for two hyperparameters in equation (1)? Is this loss used in conjunction with another loss, and if not, it appears that one hyperparameter is enough?
* It appears that Table 10 was not referred to in the text, can authors provide more information about it?

---

> ### Author Response · Authors · 2024-11-21
> **Response to Reviewer 9HJ7 Part I**
>
> Thank you for your helpful feedback! We appreciate that you found our idea of a common foundation model for various physiological signals to be interesting, and that you recognized the depth of our investigations into NormWear’s components, feature visualizations, and fusion techniques. Below, we briefly respond to each of your concerns. If you have further suggestions after reading our responses, we would be grateful to hear them and work to improve our submission accordingly. Our response has three sections.
>
> # Section 1: Response to Questions
>
> ### Question 1: What is the justification for two hyperparameters in equation (1)? Is this loss used in conjunction with another loss, and if not, it appears that one hyperparameter is enough?
> Thank you for your question. The two hyperparameters in Equation (1) correspond to the weighted average of two complementary objectives: cosine similarity to align the direction of the two representations, and Manhattan distance to bring their magnitudes closer. Although using one or two hyperparameters is theoretically equivalent, we choose the latter for its simpler implementation. By intuition, the two loss components are equivalently important, and the purpose of optimizing these two losses does not contradict each other, thus, we choose `[0.5, 0.5]` for the loss weights.
>
> ### Question 2: It appears that Table 10 was not referred to in the text, can authors provide more information about it?
> Thank you for pointing this out. We apologize for the confusion. Table 10 contains results from a simple test of the model's ability to run on edge devices and its hardware requirements. This was not a key focus of the paper but was included as a reference, as we plan future deployment and wanted to provide some initial insights for developers who may consider deploying the model. We acknowledge that this aspect is peripheral to the main focus of our work, and we will clarify this point further in the manuscript to avoid any misunderstanding.
>
> # Section 2: Response to the Concerns Raised in Weakness
>
> ## Weakness1:
> The number of subjects in the pre-training and evaluation datasets makes the conclusions intransferrable to large datasets for claims of “NormWear as a foundation model”. A foundation model is really a generalist model that can perform well on a variety of corner cases and downstream applications. Some modalities (e.g., EEG) have less than 50 pre-training/evaluation subjects, for example, their evaluation of “Driver Fatigue detection” has only 12 × 20% = [2–3] subjects in the test set, which is very low to conclude generalizable performance and conclusions for health applications. I believe this weakens the conclusion of NormWear being “[the first] foundation model specifically designed for wearable sensing data, capable of processing any number of multivariate signals from sources such as the heart, skin, brain, and physical body.” I recommend the author revise the language or provide additional empirical backup for NormWear being a foundation model.
>
> **Response1:** We sincerely thank the reviewer for highlighting the critical importance of dataset size and diversity in supporting the claim of NormWear as a foundation model. We deeply appreciate this thoughtful suggestion, as it has guided us to further strengthen the empirical foundation of our work.
>
> To address this concern, we have significantly expanded both the pretraining and downstream evaluation datasets to ensures a more comprehensive evaluation across diverse health-related tasks.:
>
> ***(a) Pretraining Data***
> Originally, for computational efficiency, we had not utilized all available samples in our pre-training dataset. To address this, we have now included all available samples and supplemented the dataset with several additional collections, including datasets containing both ECG and PPG signals, as well as a large multi-channel EEG dataset with over 50,000 samples. As a result:
> - The pre-training dataset has grown from **37k samples (62 hours)** to **241k samples (402 hours)**.
> - Leveraging time series augmentation techniques detailed in the appendix, we further expanded the dataset to **2.5 million samples (approximately 4000 hours)**.
>
> ***(b) Downstream Data***
> We incorporated several new datasets to further validate the model's performance. These include:
> - An EEG dataset with **500 subjects and five tasks**.
> - A dataset focused on muscular disorders.
> - A new ECG abnormality detection dataset.
>
> After running all experiments with these expanded datasets and additional tasks, our results confirm that **NormWear continues to achieve peak performance across a wide range of scenarios**. We have updated the relevant tables in the manuscript to reflect these findings, ensuring they provide a clear and robust empirical basis for our claims. We hope these updates satisfactorily address the reviewer's concerns and further substantiate NormWear's position as a foundation model for wearable sensing data.

---

> ### Author Response · Authors · 2024-11-21
> **Response to Reviewer 9HJ7 Part II**
>
> ## Weakness 2:
> Several major claims in the paper seem overstated. For example, the delta between NormWear and Chronos in Table 1 seems very small considering that Chronos is not even a proper foundation model on physiological signals (Chronos is just a model trained on some time-series datasets, and to the best of my knowledge, there’s no prior work showing that Chronos is even close to SOTA for physiological signals such as PPG/ECG/EEG). Despite this shortcoming for Chronos, its difference between NormWear in the first 8 evaluations is very small, and in some cases it is even better. Similarly, authors make several big claims about processing frequency domain and CWT (see examples above), however, in Table 9, they show that the difference between processing with CWT vs. raw input is not that much (76.25 vs. 78.27). I recommend authors provide further explanation/discussion regarding these claims.
>
> **Response 2:** We sincerely thank the reviewer for highlighting the need for further discussion and context regarding our claims and evaluation. This feedback has been instrumental in refining our manuscript.
>
> **1) New baselines:** We have incorporated statistical baselines, CLAP, and TF-C in our evaluation to provide a more comprehensive comparison. These additional baselines offer clearer context for understanding our model's performance. Results, including comparisons with these baselines, have been updated in the manuscript as well. To further clarify our choices: modeling multivariate wearable signals, which include a wide range of modalities such as heart, brain, and physical motion signals, poses unique challenges, as no universally recognized open-source baseline or state-of-the-art model exists in this domain. To address this, we selected baselines from three representative paradigms:
>
> - **Statistical features:** Handcrafted statistical features are a traditional yet widely used method in signal processing literature. We included this baseline as a sanity check to benchmark our model against simpler approaches.
> - **SSL Time-series frameworks:** Wearable sensory data is inherently sequential, making time-series modeling strategies relevant. We compared our approach with Chronos, a state-of-the-art time-series forecasting framework, and TF-C, a commonly used self-supervised learning method for time-series data.
> - **Spectrogram-based methods:** Spectrogram-based modeling has demonstrated state-of-the-art performance in tasks such as music classification and physiological signal analysis. CLAP was selected as a representative baseline for this paradigm.
>
> **2) Linear Probing Algorithm:** To ensure better reproducibility, numerical stability, and fairness in performance comparison, we have updated our linear probe implementation. Previously, our results were based on a custom linear layer optimized with the Adam optimizer. To standardize the evaluation pipeline, we now utilize well-established methods for solving the linear probe tasks. Specifically, classification tasks are addressed using Newton's method with conjugate gradient, while regression tasks (e.g., vital signs prediction) are solved using Cholesky decomposition with a closed-form solution. These updates provide a more robust and interpretable framework for evaluating our model's representations across various tasks.
>
> **3) CWT vs. raw input:** We recognize the importance of providing a deeper discussion regarding the differences observed. While the performance gap between CWT and raw time-series input might appear modest (76.25 vs. 78.27), this outcome underscores the robustness of our model. Specifically:
> - The results demonstrate that our model is capable of effectively extracting meaningful representations regardless of input type, which is a key strength of NormWear.
> - The ablation study confirms that using CWT does indeed yield better results, validating its utility in certain scenarios.
> We have revised the manuscript to emphasize these points, ensuring that our claims are well-supported and balanced. The updated main result table is attached below, which is also updated under Experiment section. Thank you again for this valuable feedback, which has helped us provide a clearer and more nuanced discussion.
>
> The new result table is updated in the next part of the response, and we've also updated it in the manuscript under Experiment section.

---

> ### Author Response · Authors · 2024-11-21
> **Response to Reviewer 9HJ7 Part III**
>
> ## Weakness 2 - Continued
>
> | **Downstream Tasks**                              | **Statistical** | **Chronos** | **CLAP** | **TF-C** | **NormWear (Ours)** |
> |---------------------------------------------------|-----------------|-------------|----------|----------|---------------------|
> | WESAD                                             | 66.213          | 71.489      | 72.383   | 69.865   | **76.060**          |
> | UCI-HAR                                           | 95.784          | 91.593      | 96.420   | 96.892   | **98.954**          |
> | DriverFatigue                                     | 63.249          | **76.722**  | 61.889   | 66.882   | 74.292              |
> | **Activity Recognition Avg.**                     | 75.082          | 79.935      | 76.897   | 77.880   | **83.102**          |
> |---------------------------------------------------|-----------------|-------------|----------|----------|---------------------|
> | Epilepsy (eye open)                               | 82.489          | 82.41       | 85.094   | 89.153   | **92.743**          |
> | Epilepsy (eye close)                              | 87.457          | 88.218      | 89.867   | 94.416   | 94.828              |
> | Epilepsy (health area)                            | 86.274          | 81.08       | 83.711   | 85.619   | **88.541**          |
> | Epilepsy (tumor area)                             | 82.816          | 81.034      | 83.644   | 86.348   | 87.197              |
> | Epilepsy (seizure)                                | 88.272          | 97.572      | **97.734**| 93.998   | 97.053              |
> | GAMEEMO                                           | 51.009          | 53.747      | 52.551   | **56.275**| 54.937              |
> | **EEG Main Tasks Avg.**                           | 79.720          | 80.677      | 82.100   | 84.302   | **85.883**          |
> |---------------------------------------------------|-----------------|-------------|----------|----------|---------------------|
> | PhysioNet Challenge                               | 42.592          | 48.166      | 43.058   | **50.934**| 46.904              |
> | ECG-Abnormal                                      | 97.092          | 98.585      | 97.23    | 98.275   | **99.140**          |
> | PPG-BP (HTN)                                      | 59.499          | 52.425      | 56.757   | **65.229**| 62.341              |
> | PPG-BP (DM)                                       | 47.823          | 51.164      | 42.455   | **57.883**| 55.893              |
> | PPG-BP (CVA)                                      | **71.25**       | 50.278      | 51.667   | 58.125   | 70.625              |
> | PPG-BP (CVD)                                      | 51.219          | 58.31       | 50.91    | **58.674**| 51.773              |
> | PhysioNet EMG                                     | **99.309**      | 61.6        | 98.627   | 78.308   | 99.216              |
> | **Risk Evaluation Avg.**                          | 66.969          | 60.075      | 62.958   | 66.775   | **69.413**          |
> |---------------------------------------------------|-----------------|-------------|----------|----------|---------------------|
> | Noninvasive-BP                                    | 92.31           | 91.79       | 91.922   | 87.481   | **92.420**          |
> | PPG-Hgb                                           | 94.219          | **95.005**  | 94.291   | 93.408   | 94.632              |
> | Fetal-fPCG                                        | 98.929          | 99.048      | **99.195**| 99.077   | 99.072              |
> | **Vital Signs Avg.**                              | 95.153          | 95.281      | 95.136   | 93.322   | **95.375**          |
> |---------------------------------------------------|-----------------|-------------|----------|----------|---------------------|
> | **Micro Avg.**                                    | 76.727          | 75.276      | 76.284   | 78.255   | **80.875**          |
> | **Macro Avg.**                                    | 79.231          | 78.992      | 79.273   | 80.570   | **83.443**          |

---

> ### Author Response · Authors · 2024-11-21
> **Response to Reviewer 9HJ7 Part IV**
>
> # Section 3: Response to the suggestions on writing improvements:
>
> Many important details of technical implementation is missing from the paper, I recommend the authors incorporate all necessary information to aid the reader. I provide few examples below:
> - **Q1:** Information about how patches are selected and how many patches are there for each segment, appear to be missing.
>
> - **Q2:** Architectural hyperparameters regarding the tokenizer, the reconstruction module (de-tokenization), the details of the encoder/decoder transformer (token dimension, number of attention heads, positional encoding, dimension of MLP hidden layer, normalization, ...) appear to be missing.
>
>  **A1&2:** We've updated these technical details in appendix A table 8.
>
> - **Q3:** The details regarding the downstream evaluations (linear probing) appear to missing
>
>  **A3:** We've updated the linear probing pipeline, the details are stated in previous response section.
>
> - **Q4:** The details about how sentences are chosen in Section 3.2, what language model (or encoder) was used to get the “question semantic” embeddings appear to be missing. Hyperparameters of equation 1/2 and L295-311 appear to be missing from the paper.
>
>  **A4:** We've add the citation for the text encoder we used, as well as sentence template example under appendix B. For the hyperparmeter of the loss function, we've updated it in appendix A table 6.
>
> - **Q5:** Details of masking strategies in Table 8 are missing. We did explain with text and figures.
> It would be great if the authors provide details about how confidence bounds are selected in Tables, e.g., Table 1. It is surprising that they get such narrow confidence bounds with such small N (e.g., 2/3 for Driver Fatigue detection if I understand correctly)?
>
>  **A5:** Thank you for pointing these out. The reported score was calculated from repeated running sessions without setting a fixed random seed. Though in the updated table we update the linear prob algorithm to be more deterministic and reproducible with Newton’s method for classification and Cholesky’s method for regression.
>
> Regarding the presentation improvement, we truly appreciate the time and effort you have taken to provide insights on how to improve the manuscript. We understand that clarity and flow are essential. To address the concerns you raised, we have made significant revisions to the paper, not only improving the writing but also incorporating updates to the dataset size, new baselines, and results. These changes have naturally resolved many of the earlier writing issues, while others were directly addressed in our revisions. We believe these updates enhance both the clarity of our claims and the overall robustness of our findings.
>
> As some of the specific sentences you mentioned have already been reworded or removed in the revised version, and given that the modifications are substantial, we are unable to list each change individually here. However, we sincerely appreciate the points you raised, as they have significantly contributed to improving the overall quality and clarity of the presentation. If you have any additional concerns or areas where clarity could be further improved, we would be grateful for your further suggestion. Thank you again for your constructive feedback!

---

> > ### Comment · Reviewer_9HJ7 · 2024-11-23
> > **Response to authors**
> >
> > Dear authors, I really appreciate and command your efforts for addressing my comments. Although I appreciate the improvements to the manuscript and the size of the datasets and the general idea of NormWear, my main concerns still remain the same and I don't think the manuscript, at its current form, is ready for its full potential.
> >
> > My main concerns are still overstated claims of "foundation model for multivariate physiological signals" despite the data for pre-training and (some) evaluation tasks being extremely small to call NormWear a foundation model for a multitude of physiological signals. For example, total amount of waveform data used for pre-training in this study is 400 hours, which is arguably by orders of magnitude smaller than the standards for even a single waveform modality. Additionally, the paper presents main results and ablation experiments on different amount of data, which makes the learnings/choices not really translatable. I totally understand that it is hard to squeeze in experiments in short period of rebuttal time, but this manuscript at its current form is not ready for acceptance. Therefore, sadly, I will have to maintain my score.

---

> ### Author Response · Authors · 2024-11-27
> **Response to Reviewer 9HJ7**
>
> Dear reviewer,
>
> Thank you for your timely reply and your constructive feedback.
>
> **Foundation model for multimodal wearable-based physiological signals:** Modern wearables come in diverse form factors and capture a wide range of modalities, from brain activity and cardiac or respiratory function to physical movements. While recent advances in AI/ML have leveraged wearable signals, no single foundation model exists that is agnostic to input modality, effectively represents diverse combinations of wearable modalities, and addresses a variety of health sensing tasks.
>
> There exists closed source models or data, or focus entirely on designing a specific deep learning based method on one single application. This issue in the field has to be acknowledged and it requires a more generalized modeling methodology. In the current work, **instead of focusing on training our model on a single large dataset we focus our attention on diversifying datasets containing different wearable input modalities and sensing tasks.** We conduct all of our experiments on **entirely publicly available datasets,** and we do ensure to release all the **code, cleaned preprocessed data, and model checkpoints** generated during this study for the core benefit of the community for future research.
>
> **Problem addressed by our method:** In our contribution, we address most of the core problems, where NormWear can take any wearable sensor signals (for heart, brain, IMU, muscle, PCG, etc.) with **any number of channels** (e.g. PPG only, PPG+ECG, IMU+PPG, 1/4/12/64 channels of EEG, etc.), with **any input channel order,** and shows its informative representation can be easily applied on **diverse health related tasks: for mental health, activities recognition, brain tumor area detection, seizure detection, muscle dystrophies and neuropathies detection, and several other blood related disease risk evaluation with common vital sign estimation tasks.** We are glad to see reviewers’ acknowledgement of our core contributions. The applicability of our model on **both large and small downstream datasets perfectly matches the reviewer’s belief that “A foundation model is really a generalist model that can perform well on a variety of corner cases and downstream applications.”**
>
> **Role of pretraining size:** We acknowledge that pretraining size is undeniably important to ensure the foundation model’s generalizability on various downstream tasks. Nevertheless, this single aspect should not overshadow the importance of addressing core modeling challenges, engineering considerations, and domain-specific issues. In our work, we emphasize the need to prioritize these foundational strategies rather than solely focusing on pretraining dataset size. Our approach not only demonstrates the potential of scaling with 2.5 million signal samples with varied channel configurations but also underscores the value of tackling fundamental modeling challenges to ensure meaningful progress in the field.
>
> To this end, we sincerely thank the reviewer once again for their valuable feedback and guidance. The insights have not only significantly helped to improve our work but also will leave valuable guidance on OpenReview, serving as a valuable reference for future researchers.
>
> We would like to thank you for the opportunities to discuss and contribute to the continued advancement of the field.
>
> Best,
> Authors

---

> ### Author Response · Authors · 2024-12-04
> **Response to Reviewer 9HJ7**
>
> ## Regarding the pretrain data size:
> We appreciate the reviewer's feedback and would like to clarify some details regarding the size and composition of our pretraining dataset. Specifically, our dataset consists of **2,576,418 segment samples, totaling approximately 4,294 hours**. This figure differs from the 400 hours mentioned in the review. Additionally, these segments encompass a variety of sensor channels, which we believe challenges the statement that “this is arguably by orders of magnitude smaller than the standards for even a single waveform modality.” When treating the sensors individually (e.g., EEG, PPG, IMU, ECG, GSR, etc.), our dataset contains **8,965,538 sensor signal series, totaling 14,943 hours**.
>
> To further clarify and enhance transparency, we have included two pie charts in the overview figure, illustrating the proportions of different sensor types and physiological signals represented in our dataset. These visualizations aim to provide a clearer picture of the data's diversity and breadth.
>
> That said, while we acknowledge the importance of dataset size, we maintain the perspective stated in our previous response: the pretraining dataset size is one aspect of our work, but it should not overshadow our core contributions. These include, which are also gratefully acknowledged by the reviewers, the **novel modeling strategy accommodating flexible numbers and types of sensor channels, the peak performance achieved across diverse health applications, the first to achieve zero-shot inference on multi-variate wearable signals, and the fact that our entire work is open-sourced for the community**. We hope this provides a more comprehensive view of our contributions and addresses the concerns raised.
>
> ## Regarding the ablation study alignment:
> We are pleased to share the up-to-date version of our ablation study, aligned with the pipeline update as described in our 1st round of response. We've included all of the core ablations including the data input format, different masking scheme and different inter-channel fusion logic. All the experiments are now completed with same amount of data (both pretrain and downstream), and all observations and arguments remain consistent, and the detailed result are presented in the following table (in next response) which we will also update in the Appendix as well.

---

> > ### Author Response · Authors · 2024-12-04
> > **Continue Response to Reviewer 9HJ7**
> >
> > | Metric                     | NormWear (Final) | Masking Unstructured | Masking Time only | Masking Scale only | Raw time series | w/o fusion | Cross attn. fusion | Mean fusion |
> > |----------------------------|------------------|-----------------------|-------------------|--------------------|-----------------|-----------|--------------------|-------------|
> > | WESAD                      | 76.06           | 71.46                | 71.952            | 72.201             | 70.862          | 72.209    | 74.165             | 71.99       |
> > | UCI-HAR                    | 98.954          | 97.097               | 98.438            | 98.106             | 97.969          | 97.793    | 96.908             | 97.566      |
> > | DriverFatigue              | 74.292          | 72.719               | 73.424            | 78.354             | 73.854          | 73.252    | 60.308             | 72.552      |
> > | Activity Recognition Avg. | **83.102**          | 80.425               | 81.271            | 81.342             | 80.895          | 81.085    | 77.127             | 80.703      |
> > | GAMEEMO                    | 54.937          | 58.043               | 56.77             | 55.771             | 54.651          | 57.695    | 56.724             | 58.079      |
> > | Epilepsy (eye open)        | 92.743          | 89.521               | 91.895            | 89.407             | 91.978          | 90.966    | 84.075             | 89.817      |
> > | Epilepsy (eye close)       | 94.828          | 93.471               | 94.808            | 93.786             | 94.781          | 94.399    | 93.589             | 93.912      |
> > | Epilepsy (health area)     | 88.541          | 86.812               | 88.51             | 87.317             | 88.045          | 87.866    | 86.899             | 87.248      |
> > | Epilepsy (tumor area)      | 87.197          | 86.524               | 88.254            | 85.502             | 85.619          | 86.599    | 86.861             | 87.152      |
> > | Epilepsy (seizure)         | 97.053          | 96.59                | 97.791            | 95.29              | 97.722          | 97.477    | 96.351             | 96.719      |
> > | Neuron State Recognition Avg. | 85.883      | 85.160               | **86.338**            | 84.512             | 85.466          | 85.834    | 84.083             | 85.488      |
> > | Noninvasive-BP             | 92.42           | 90.124               | 90.65             | 91.152             | 89.85           | 88.356    | 92.759             | 88.719      |
> > | PPG-Hgb                    | 94.632          | 95.314               | 95.055            | 94.713             | 93.832          | 95.031    | 93.413             | 95.086      |
> > | Fetal-fPCG                 | 99.072          | 98.63                | 99.121            | 98.926             | 98.977          | 98.582    | 99.145             | 98.771      |
> > | Vital Sign Avg.            | **95.375**          | 94.689               | 94.942            | 94.93              | 94.22           | 93.99     | 95.106             | 94.192      |
> > | PPG-BP (HTN)               | 62.341          | 58.88                | 55.333            | 59.23              | 52.614          | 61.85     | 60.983             | 63.577      |
> > | PPG-BP (DM)                | 55.893          | 61.074               | 48.386            | 58.896             | 62.012          | 58.333    | 62.8               | 62.2        |
> > | PPG-BP (CVA)               | 70.625          | 56.389               | 58.472            | 64.167             | 56.181          | 61.319    | 61.458             | 59.236      |
> > | PPG-BP (CVD)               | 51.773          | 52.572               | 46.557            | 55.666             | 54.812          | 48.417    | 53.585             | 46.961      |
> > | ECG-Abnormal               | 99.14           | 99.085               | 99.316            | 98.296             | 97.701          | 99.429    | 99.441             | 99.268      |
> > | PhysioNet EMG              | 99.216          | 85.16                | 95.49             | 83.922             | 93.756          | 93.715    | 95.49              | 86.749      |
> > | Risk Evaluation Avg.       | **73.165**          | 68.86                | 67.259            | 70.0295            | 69.513          | 70.5105   | 72.294             | 69.665      |
> > | Micro Avg.                 | **82.762**    | 80.526               | 80.568            | 81.150             | 80.845          | 81.294    | 80.831             | 80.867      |
> > | Macro Avg.                 | **84.381**        | 82.284               | 82.453            | 83.090             | 82.523          | 82.855    | 82.152             | 82.512      |
> >
> > We hope our additional supplementary results and the clarification could better address the reviewer's concern, and we thank you again for your constructive feedback.

---

### Official Review · Reviewer_ZArW · 2024-10-26

**Soundness:** 1
**Presentation:** 1
**Contribution:** 3
**Rating:** 3
**Confidence:** 4

**Summary:**

The paper introduces NORMWEAR, a foundation model designed to address challenges in processing multi-modal wearable sensing data for healthcare applications. NORMWEAR can process diverse physiological signals, such as ECG, EEG, and PPG, by leveraging an innovative tokenization approach and a channel-aware attention mechanism, which together enable it to handle multivariate data efficiently. Additionally, NORMWEAR is capable of zero-shot inference, achieved through a representation alignment technique. This allows the model to interpret and apply its insights to new health-related applications without needing to be retrained, making it highly adaptable to different contexts. Through an evaluation across 12 different health downstream tasks, NORMWEAR has shown improvements over baseline time-series models.

**Strengths:**

* The fusion approach is interesting and well studied. There are multiple ablations showing the strength of each of these approaches, and they are well thought through and explained. This addresses an important problem within the space, in which the type and # of modalities will be inconsistent.
* The tokenization approach is interesting and well-justified, using prior work from the biosignal space to justify each of the steps. This addresses an critical problem within our space to identify better time-series tokenization methods. Further experimentation and ablations on the tokenization method would have been appreciated though.
* Good results on many datasets. The multitude of very different tasks shows a generalization zero-shot performance of the model, which is quite notable.

**Weaknesses:**

* Semantic alignment training procedure is unclear
	* After pre-training the backbone encoder, the embedding space is aligned with the semantic text. However, I cannot seem to understand how this is done nor what datasets are used (i.e. are they the same as the pre-training datasets?). This seems to be a critical aspect of the model in order to enable zero-shot performance, but little detail is given. Two papers are cited, Zhang et al., 2024; Liu et al., 2024, but they do not seem to imply one specific approach.
* Insufficient baselines
	* It is argued that chronos is a SOTA method, however, as noted, chronos was designed for time-series forecasting, which none of the downstream tasks are.
	* The ViT baseline is not well explained in how it is set up to do zero-shot downstream, including it's learning objective.
	* Ideally, the baselines should encompass the SOTA method for a given task, so as to understand how this model compares against each task specifically. The scope of the paper is "towards" a foundation model so this isn't a hard requirement, but would be nice.
* Many model components in the Memory Stream Inspired Mechanism in Sec 2.4 are not clearly explained
	* In MSiTF, it is argued that the representations are optimized for human sensing, how does this occur, specifically? This is not justified clearly. In Fig. 3, standard deviation and mean are used, but this does not seem to be explained in this section.
	* How is recency score important? It is stated that the further the time-step to the most recent time step, the lower te score, what time steps are being considered here? It is not clear, especially because the query is text and key/values are embedding time-series, and thus on different time scales.
	* Importancy score seems to act as a gate for the inputs time stamps, but it is not explained how the gate determined to be on or off.
	* The text states that final score is a summation, but how are scores used? According to Fig. 3., it looks like the scores operate independently from each other in different model components, rather than being summed together.
	* In Eq. 1, it seems like there are two loss components, a l1 loss and cosine distance. Why are they both used together when they both work towards increasing similarity between Y and \hat{Y}? There is no text nor empirical results justifying this.
* Experimental results are somewhat lacking
	* No experimental results showing the strength of their tokenization method compared to a simple Conv1d tokenizer.
	* Only having one metric reported makes it difficult for us to understand whether the performance gain is consistent.
	* In Fig. 4a), t-SNE clusters of the different classes being different is not too surprising, as each signal is quite different, so it would be nice to understand visualize and understand how the model is able to capture differences among specific classes or specific risk levels.
	* In Fig. 4b), it is unclear what the visualizations extracted by the intermediate layer imply and how they show that the model has learned meaningful information.
* Lack of a prior work section makes it very difficult for readers to understand where this work sits within the greater foundation model space.
* Figures are hard to follow. Some ideas are introduced in the figures, but do not seem to be explained in the main text.
	* In Fig. 3, Mean/Standard Deviation + Likelihood Parameters are not explained in the main text.

**Questions:**

Please seek weaknesses above.

---

> ### Author Response · Authors · 2024-11-21
> **Response to Reviewer ZArW Part I**
>
> Thank you for your helpful feedback! We appreciate that you found our fusion and tokenization approaches to be well-studied and justified, and that you recognized the strength of our ablations and the impressive generalization performance of NORMWEAR across diverse tasks. Below, we briefly respond to each of your concerns. If, after reading our responses, you still have suggestions for improvement, we would love to hear them and further refine our work.
>
> ## Weakness 1: Semantic alignment training procedure is unclear
> - After pre-training the backbone encoder, the embedding space is aligned with the semantic text. However, I cannot seem to understand how this is done nor what datasets are used (i.e. are they the same as the pre-training datasets?). This seems to be a critical aspect of the model in order to enable zero-shot performance, but little detail is given. Two papers are cited, Zhang et al., 2024; Liu et al., 2024, but they do not seem to imply one specific approach.
>
> **Response**
>
> Thank you for your valuable comment and for pointing out the need for clarification on the semantic alignment training procedure.
>
> As described in Section 2.5 of the manuscript, we propose a novel temporal fusion mechanism that aligns the backbone encoder's output with a semantic space, enabling zero-shot inference. Specifically, we use the semantic embedding of a query sentence and the latent output from the backbone model to create a fused representation. The alignment is supervised using a loss function that incorporates both Manhattan distance and cosine similarity, ensuring that the physiological representation is aligned with the semantic representation in both direction and magnitude.
>
> The pre-training is still conducted on the pre-training datasets, and the semantic alignment step occurs after the backbone encoder has been trained. The pre-training and downstream datasets are always disjoint throughout all experiments in this paper, meaning that the downstream evaluation is conducted on data that the model has not seen during pre-training.
>
> We've updated the above clarification in the paper under section 2.4. We hope this clarifies the training procedure. Thank you for your insightful feedback, which helped improve the explanation.
>
> ## Weakness 2: Insufficient baselines
> - It is argued that chronos is a SOTA method, however, as noted, chronos was designed for time-series forecasting, which none of the downstream tasks are.
> - The ViT baseline is not well explained in how it is set up to do zero-shot downstream, including it's learning objective.
> - Ideally, the baselines should encompass the SOTA method for a given task, so as to understand how this model compares against each task specifically. The scope of the paper is "towards" a foundation model so this isn't a hard requirement, but would be nice.
>
> **Response**
>
> Thank you for your thoughtful feedback regarding the baselines used in our work. We appreciate your insight into the need for more targeted baselines, particularly for zero-shot performance and task-specific comparisons.
> We have now added additional baselines to better address this concern.
>
> Regarding the ViT baseline, we would like to clarify that it is not used for zero-shot downstream tasks, but rather as a "vision-based" approach for comparison. We understand that a spectrum-based model would provide a more relevant baseline for our tasks, and we have now included this in the updated baselines. The following are the details of our updates:
>
> **1) New baselines:** We have incorporated statistical baselines, CLAP, and TF-C in our evaluation to provide a more comprehensive comparison. These additional baselines offer clearer context for understanding our model's performance. Results, including comparisons with these baselines, have been updated in the manuscript as well. To further clarify our choices: modeling multivariate wearable signals, which include a wide range of modalities such as heart, brain, and physical motion signals, poses unique challenges, as no universally recognized open-source baseline or state-of-the-art model exists in this domain. To address this, we selected baselines from three representative paradigms:
>
> - **Statistical features:** Handcrafted statistical features are a traditional yet widely used method in signal processing literature. We included this baseline as a sanity check to benchmark our model against simpler approaches.
> - **SSL Time-series frameworks:** Wearable sensory data is inherently sequential, making time-series modeling strategies relevant. We compared our approach with Chronos, a state-of-the-art time-series forecasting framework, and TF-C, a commonly used self-supervised learning method for time-series data.
> - **Spectrogram-based methods:** Spectrogram-based modeling has demonstrated state-of-the-art performance in tasks such as music classification and physiological signal analysis. CLAP was selected as a representative baseline for this paradigm.

---

> ### Author Response · Authors · 2024-11-21
> **Response to Reviewer ZArW Part II**
>
> ## Weakness 2: Insufficient Baseline - Continued
> **2) Linear Probing Algorithm:** To ensure better reproducibility, numerical stability, and fairness in performance comparison, we have updated our linear probe implementation. Previously, our results were based on a custom linear layer optimized with the Adam optimizer. To standardize the evaluation pipeline, we now utilize well-established methods for solving the linear probe tasks. Specifically, classification tasks are addressed using Newton's method with conjugate gradient, while regression tasks (e.g., vital signs prediction) are solved using Cholesky decomposition with a closed-form solution. These updates provide a more robust and interpretable framework for evaluating our model's representations across various tasks.
>
> New results, including comparisons with these baselines is attached below, and we've also updated it in the manuscript as well.
>
> | **Downstream Tasks**                              | **Statistical** | **Chronos** | **CLAP** | **TF-C** | **NormWear (Ours)** |
> |---------------------------------------------------|-----------------|-------------|----------|----------|---------------------|
> | WESAD                                             | 66.213          | 71.489      | 72.383   | 69.865   | **76.060**          |
> | UCI-HAR                                           | 95.784          | 91.593      | 96.420   | 96.892   | **98.954**          |
> | DriverFatigue                                     | 63.249          | **76.722**  | 61.889   | 66.882   | 74.292              |
> | **Activity Recognition Avg.**                     | 75.082          | 79.935      | 76.897   | 77.880   | **83.102**          |
> |---------------------------------------------------|-----------------|-------------|----------|----------|---------------------|
> | Epilepsy (eye open)                               | 82.489          | 82.41       | 85.094   | 89.153   | **92.743**          |
> | Epilepsy (eye close)                              | 87.457          | 88.218      | 89.867   | 94.416   | 94.828              |
> | Epilepsy (health area)                            | 86.274          | 81.08       | 83.711   | 85.619   | **88.541**          |
> | Epilepsy (tumor area)                             | 82.816          | 81.034      | 83.644   | 86.348   | 87.197              |
> | Epilepsy (seizure)                                | 88.272          | 97.572      | **97.734**| 93.998   | 97.053              |
> | GAMEEMO                                           | 51.009          | 53.747      | 52.551   | **56.275**| 54.937              |
> | **EEG Main Tasks Avg.**                           | 79.720          | 80.677      | 82.100   | 84.302   | **85.883**          |
> |---------------------------------------------------|-----------------|-------------|----------|----------|---------------------|
> | PhysioNet Challenge                               | 42.592          | 48.166      | 43.058   | **50.934**| 46.904              |
> | ECG-Abnormal                                      | 97.092          | 98.585      | 97.23    | 98.275   | **99.140**          |
> | PPG-BP (HTN)                                      | 59.499          | 52.425      | 56.757   | **65.229**| 62.341              |
> | PPG-BP (DM)                                       | 47.823          | 51.164      | 42.455   | **57.883**| 55.893              |
> | PPG-BP (CVA)                                      | **71.25**       | 50.278      | 51.667   | 58.125   | 70.625              |
> | PPG-BP (CVD)                                      | 51.219          | 58.31       | 50.91    | **58.674**| 51.773              |
> | PhysioNet EMG                                     | **99.309**      | 61.6        | 98.627   | 78.308   | 99.216              |
> | **Risk Evaluation Avg.**                          | 66.969          | 60.075      | 62.958   | 66.775   | **69.413**          |
> |---------------------------------------------------|-----------------|-------------|----------|----------|---------------------|
> | Noninvasive-BP                                    | 92.31           | 91.79       | 91.922   | 87.481   | **92.420**          |
> | PPG-Hgb                                           | 94.219          | **95.005**  | 94.291   | 93.408   | 94.632              |
> | Fetal-fPCG                                        | 98.929          | 99.048      | **99.195**| 99.077   | 99.072              |
> | **Vital Signs Avg.**                              | 95.153          | 95.281      | 95.136   | 93.322   | **95.375**          |
> |---------------------------------------------------|-----------------|-------------|----------|----------|---------------------|
> | **Micro Avg.**                                    | 76.727          | 75.276      | 76.284   | 78.255   | **80.875**          |
> | **Macro Avg.**                                    | 79.231          | 78.992      | 79.273   | 80.570   | **83.443**          |

---

> ### Author Response · Authors · 2024-11-21
> **Response to Reviewer ZArW Part III**
>
> ## Weakness 3: Many model components in the Memory Stream Inspired Mechanism in Sec 2.4 are not clearly explained
> - In MSiTF, it is argued that the representations are optimized for human sensing, how does this occur, specifically? This is not justified clearly. In Fig. 3, standard deviation and mean are used, but this does not seem to be explained in this section.
> - How is recency score important? It is stated that the further the time-step to the most recent time step, the lower te score, what time steps are being considered here? It is not clear, especially because the query is text and key/values are embedding time-series, and thus on different time scales.
> - Importancy score seems to act as a gate for the inputs time stamps, but it is not explained how the gate determined to be on or off.
> - The text states that final score is a summation, but how are scores used? - According to Fig. 3., it looks like the scores operate independently from each other in different model components, rather than being summed together.
> - In Eq. 1, it seems like there are two loss components, a l1 loss and cosine distance. Why are they both used together when they both work towards increasing similarity between Y and \hat{Y}? There is no text nor empirical results justifying this.
>
> **Response**
> Thank you for your detailed feedback regarding the Memory Stream Inspired Mechanism (MSiTF) and the components involved in our temporal fusion process. We've updated the text in section 2.5 for better clarification. We appreciate your suggestions, and we put a summerized clarifications below to address the points you've raised.
>
> **1) Optimization for Human Sensing:**
> We understand the importance of clearly explaining how the model is optimized for human sensing tasks. In our approach, we leverage a temporal fusion mechanism that incorporates recency scores, relevance scores, and importance scores to prioritize specific time steps based on their relevance to the task at hand. The recency score is derived from the temporal distance of each patch to the most recent time step, with closer time steps receiving higher weights due to their stronger correlation with immediate human actions or conditions. This is particularly important in human sensing tasks, where recent data points are often more critical for accurate prediction.
>
> **2) Use of Mean and Standard Deviation:**
> To address the concern about the use of mean and standard deviation, we clarify that this is part of our variational-inspired approach. By computing the mean and standard deviation of patch embeddings, we inject stochasticity into the representation process, encouraging the model to explore variations in the data. This mechanism allows the model to capture the inherent variability in human sensing data, which is essential for robust zero-shot inference. And from the result table we observe that the performance boost up with this variational mechanism.
>
> **3) Importance Score as a Gate:**
> The importance score acts as a binary gate that determines whether a specific representation at a given time step should be retained or discarded. This gate is learned during pretraining via a trainable linear transformation, which optimizes the model's ability to select the most relevant information at each time step. The importance score is derived using a Gumbel-softmax function, which allows the model to make hard decisions while remaining differentiable during training.
>
> **4) Summation of Scores:**
> Regarding the final score, it is indeed a summation of the relevance, recency, and importance scores. These scores are used to weight the contributions of each time step when aggregating the patch representations. This weighted sum allows the model to prioritize more relevant, recent, and important patches in the fusion process, leading to a more informative fixed-length representation.
>
> **5) Loss Components in Eq. 1:**
> The use of both L1 loss and cosine distance in our objective is to balance two complementary aspects of alignment: L1 loss focuses on minimizing the absolute difference between the target and predicted representations, while cosine distance encourages similarity in directionality between the two. We find that both losses contribute to a more effective alignment of the representations, enhancing the model's performance in zero-shot inference tasks.
> We hope these explanations address your concerns and provide more clarity on the functioning of the MSiTF mechanism. Thank you again for your constructive feedback, which has helped us refine our approach.

---

> ### Author Response · Authors · 2024-11-21
> **Response to Reviewer ZArW Part IV**
>
> ## Weakness 4: Experimental results are somewhat lacking
>
> - No experimental results showing the strength of their tokenization method compared to a simple Conv1d tokenizer.
> - Only having one metric reported makes it difficult for us to understand whether the performance gain is consistent.
> - In Fig. 4a), t-SNE clusters of the different classes being different is not too surprising, as each signal is quite different, so it would be nice to understand visualize and understand how the model is able to capture differences among specific classes or specific risk levels.
> - In Fig. 4b), it is unclear what the visualizations extracted by the intermediate layer imply and how they show that the model has learned meaningful information.
>
> **Response**
>
> Thank you for your valuable feedback. We appreciate the opportunity to clarify and improve our presentation of the experimental results.
>
> Regarding the comparison with the Conv1D tokenizer, we did have an explicit evaluation in Appendix Table 11, where we compare our approach to Conv1D tokenization on raw time series data. This comparison highlights the advantages of our tokenization method in terms of performance.
>
> To address the concern about the limited metric reporting, we have expanded the metrics section to include additional performance measures, such as AUC ROC, AUC PR, accuracy, precision, recall, and F1 score. These metrics provide a more comprehensive view of the model's performance and demonstrate that NormWear generally achieves peak performance across the tasks. The result table is updated in the next part of response, and we've also added it to the manuscript under Experiment section.
>
> For the t-SNE visualizations, we would like to clarify that they are not intended to emphasize the superiority of the clusters but rather to demonstrate that the model is capable of recognizing patterns without needing to be explicitly told which sensor is involved. The model's ability to distinguish these patterns and assign appropriate attention scores is evident in the visualizations. While we understand that this may seem intuitive, we wanted to showcase this as a part of our model's capabilities, without overemphasizing the significance of the t-SNE visualization.
>
> Regarding the feature analysis of the intermediate layers, we appreciate your point. Our aim with these visualizations is not to make claims about the model's learned features, but rather to offer insights into what the model captures at each layer. The nonlinear dynamics is intended to help us better understand the behavior of the model. Such as Lyapunov exponent (sensitivity to initial conditions), Hurst exponent (self-correlation/seasonality), and persistence entropy (unpredictability in system states).
> Once again, thank you for your feedback. We have updated the paper to include these clarifications and hope that these changes address your concerns effectively.

---

> ### Author Response · Authors · 2024-11-21
> **Response to Reviewer ZArW Part V**
>
> ## Weakness 4: Experimental results are somewhat lacking - Continued
>
> #### Result table of methods performance across different groups of downstream tasks.
>
> | **Task Group**            | **Methods**         | **AUC ROC** | **AUC PR** | **Accuracy** | **Precision** | **Recall** | **F1 Score** |
> |---------------------------|---------------------|-------------|------------|--------------|---------------|------------|--------------|
> | **Activity Recognition**   | Statistical         | 75.082      | 63.996     | 65.298       | 61.450        | 61.56      | 61.034       |
> |                           | Chronos             | 79.935      | 65.622     | 66.175       | 62.044        | 61.512     | 60.522       |
> |                           | CLAP                | 76.897      | 67.026     | 66.349       | 62.790        | 62.826     | 62.435       |
> |                           | TF-C                | 77.880      | 68.228     | 67.175       | 64.967        | 64.798     | 64.783       |
> |                           | **NormWear (Ours)** | **83.102**  | **76.232** | **75.254**   | **72.606**    | **72.177** | **72.053**   |
> |---------------------------|---------------------|-------------|------------|--------------|---------------|------------|--------------|
> | **EEG Main Tasks**         | Statistical         | 79.720      | 50.172     | 73.921       | 63.567        | 57.529     | 57.948       |
> |                           | Chronos             | 80.677      | 55.507     | 75.285       | 72.442        | 52.520     | 47.671       |
> |                           | CLAP                | 82.100      | 57.518     | 76.391       | 68.506        | 61.961     | 62.650       |
> |                           | TF-C                | 84.302      | 61.864     | 76.825       | 71.702        | 65.517     | 67.889       |
> |                           | **NormWear (Ours)** | **85.883**  | **66.841** | **79.182**   | **72.485**    | **69.158** | **69.698**   |
> |---------------------------|---------------------|-------------|------------|--------------|---------------|------------|--------------|
> | **Disease Risk Evaluation**| Statistical         | 66.969      | **49.256**     | 75.522       | 46.992        | 49.202     | 45.947       |
> |                           | Chronos             | 60.075      | 38.515     | 68.855       | 40.814        | 41.062     | 37.171       |
> |                           | CLAP                | 62.958      | 45.009     | 76.927       | 47.274        | 50.475     | 47.673       |
> |                           | TF-C                | 66.775      | 43.454     | 74.864       | 46.484        | 48.441     | 46.413       |
> |                           | **NormWear (Ours)** | **69.413**  | 47.683 | **77.101**   | **48.204**    | **51.841** | **49.050**   |
> |---------------------------|---------------------|-------------|------------|--------------|---------------|------------|--------------|
> | **Micro Average**          | Statistical         | 73.272      | 52.363     | 73.005       | 55.918        | 54.641     | 53.276       |
> |                           | Chronos             | 71.524      | 49.970     | 70.764       | 56.655        | 49.193     | 45.487       |
> |                           | CLAP                | 72.750      | 53.828     | 74.743       | 58.145        | 57.098     | 56.057       |
> |                           | TF-C                | 75.430      | 55.003     | 74.158       | 59.406        | 57.911     | 57.911       |
> |                           | **NormWear (Ours)** | **78.156**  | **60.220** | **77.535**   | **61.885**    | **62.148** | **61.106**   |
> |---------------------------|---------------------|-------------|------------|--------------|---------------|------------|--------------|
> | **Macro Average**          | Statistical         | 73.924      | 54.475     | 71.580       | 57.336        | 56.097     | 54.976       |
> |                           | Chronos             | 73.562      | 53.215     | 70.105       | 58.433        | 51.698     | 48.454       |
> |                           | CLAP                | 73.985      | 56.517     | 73.222       | 59.523        | 58.421     | 57.586       |
> |                           | TF-C                | 76.319      | 57.849     | 72.955       | 61.051        | 59.585     | 59.695       |
> |                           | **NormWear (Ours)** | **79.466**  | **63.585** | **77.179**   | **64.432**    | **64.392** | **63.600**   |
>
> Thank you again for your constructive feedback!

---

### Official Review · Reviewer_p9eE · 2024-11-04

**Soundness:** 3
**Presentation:** 2
**Contribution:** 3
**Rating:** 3
**Confidence:** 4

**Summary:**

This paper introduces NORMWEAR, a foundation model for multivariate wearable physiological signals. NORMWEAR uses continuous wavelet transforms and channel-aware attention to learn robust representations from diverse sensor types. Evaluated on various downstream healthcare tasks, it outperforms existing baselines. A novel fusion mechanism enables zero-shot inference for custom health applications. The authors also provide an interpretability analysis using feature visualization and nonlinear dynamics.

**Strengths:**

- The paper tackles the important and under-explored area of foundation models for multivariate wearable physiological signals. Existing foundation models for time series often struggle with the specific challenges of this data type, such as variability in patterns, frequency bands, and sensor combinations.

- The proposed NORMWEAR model incorporates thoughtful design choices, including CWT-based multi-scale representations, channel-aware attention, and a zero-shot inference mechanism. These components appear well-suited to address the complexities of wearable physiological data.

- The authors evaluate their model on a diverse set of downstream tasks spanning mental health, body state inference, biomarker estimation, and disease risk evaluation. This provides a holistic assessment of the model's capabilities.

- The paper emphasizes model interpretability through feature visualization and nonlinear dynamic analysis. This is a crucial aspect for building trust and understanding the model's behavior in healthcare applications.

- NORMWEAR demonstrates superior performance compared to Chronos, a state-of-the-art language-based foundation model, and a vision-based ViT baseline.

**Weaknesses:**

- Limited size of the pre-training dataset: The relatively small size of the pre-training dataset (around 37,000 samples with a limited number of subjects in several datasets) raises concerns about the model's ability to generalize effectively. Similar works often utilize significantly larger datasets with tens or hundreds of thousands of participants. This limitation needs to be acknowledged and addressed more thoroughly. How does this limited dataset size impact the robustness of the learned representations?

- Lack of simple statistical baselines: While comparison with Chronos and ViT provides valuable context, the inclusion of simpler statistical feature baselines would strengthen the evaluation and help establish a lower bound on performance. Also, comparisons with existing self-supervised baselines like BYOL, TF-C, or SimCLR would position this work better within the literature. This would give a clearer picture of the added value provided by the complex architecture.

- Insufficient information on dataset selection: The rationale behind the selection of pre-training and downstream datasets is not clearly articulated. Why were certain larger datasets reserved for downstream evaluation only? The paper mentions limited information on one of the datasets (BP with 1000 users) and should clarify its origin (self-collected or public). The provenance of all datasets should be explicitly provided, ideally with inline references within Tables 3 and 4.

**Questions:**

1. Impact of dataset size: Could the authors elaborate on the potential impact of the limited pre-training dataset size on the model's generalization performance? Are there plans to expand the pre-training dataset in future work?

2. Model Release: Given the lack of openly released models in this domain, I strongly encourage the authors to publicly release their model and code to facilitate future research and comparisons.

3. Tokenization terminology: The use of "tokenization" might be confusing, given its association with language models. "Pre-processing" or "feature extraction" might be more suitable terms in this context. Could the authors clarify their usage of this term?

---

> ### Author Response · Authors · 2024-11-21
> **Response to Reviewer p9eE Part I**
>
> Thank you for your detailed and insightful feedback! We appreciate that you found our work to address an important and under-explored area and that you recognized the thoughtful design of our NORMWEAR model, as well as its strong performance across diverse healthcare applications. Below, we briefly respond to each of your concerns. If, after reading our responses, you feel there are areas where our work can be further improved, we would greatly value your additional feedback to help us refine and strengthen our contributions. Our response has two sections.
>
> # Section 1: Response to the Questions
>
> ****Questions:****
>
> ****Q1**** Impact of dataset size: Could the authors elaborate on the potential impact of the limited pre-training dataset size on the model's generalization performance? Are there plans to expand the pre-training dataset in future work?
>
> ****A1**** The dataset size was indeed a critical consideration in our work. We have already expanded the pre-training dataset by 80 times with the number of samples from 37k to 2.5 million (~ 4000 hours), significantly improving its diversity and scale. A detailed explanation of this expansion and its impact on model performance is provided in the next sections of our response.
>
> ----------------------------------------------------------------------------
>
> ****Q2**** Model Release: Given the lack of openly released models in this domain, I strongly encourage the authors to publicly release their model and code to facilitate future research and comparisons.
>
> ****A2**** Thank you for your valuable suggestion and for highlighting the importance of model and code release in advancing research within this domain. We fully agree with your perspective, and we are committed to publicly releasing both our model and code upon publication. We believe this will help facilitate further research, foster transparency, and enable meaningful comparisons in future work.
>
> ----------------------------------------------------------------------------
>
> ****Q2**** Tokenization terminology: The use of "tokenization" might be confusing, given its association with language models. "Pre-processing" or "feature extraction" might be more suitable terms in this context. Could the authors clarify their usage of this term?
>
> ****A2**** Thank you for raising this concern. We acknowledge the potential for confusion with the term "tokenization" due to its association with language models. As clarified at the beginning of the tokenization section, in the context of wearable sensing, we use this term to describe the signal processing stage prior to inputting the processed data into the deep learning-based encoder. This includes the patching step, where the CWT-based scalograms are divided into smaller patches, which we refer to as “tokens”. We hope this explanation clarifies our intended usage of the term.
>
> ----------------------------------------------------------------------------
> ****Q3**** Insufficient information on dataset selection: The rationale behind the selection of pre-training and downstream datasets is not clearly articulated. Why were certain larger datasets reserved for downstream evaluation only? The paper mentions limited information on one of the datasets (BP with 1000 users) and should clarify its origin (self-collected or public). The provenance of all datasets should be explicitly provided, ideally with inline references within Tables 3 and 4.
>
> ****A3****
> Thank you for your feedback regarding dataset selection and the need for more detailed information. We have updated the manuscript to include comprehensive tables for both pre-training and downstream datasets, providing all necessary details such as citations, number of samples, types of signals, and corresponding downstream tasks where applicable. These updates tables (Table 1 and 2) are added in the main section of the paper under Method section.
>
> As for the rationale behind dataset selection:
> - For pre-training, we prioritized datasets with the largest number of samples, diverse channels, and a variety of sensors to effectively train the channel fusion module.
> - For downstream tasks, we selected datasets that cover a broad range of health-related applications, including mental health, human activity recognition (HAR), EEG-based tasks, disease risk prediction, and vital sign monitoring. This diversity ensures a comprehensive evaluation of our model across different domains.
>
> We hope these updates and clarifications address your concerns. Thank you again for highlighting these important points, which helped improve the clarity and completeness of our work.

---

> ### Author Response · Authors · 2024-11-21
> **Response to Reviewer p9eE Part II**
>
> # Section 2: Response to the concerns raised in weakness
>
> ## Weakness 1 & 2
> - Limited size of the pre-training dataset
>
> - Lack of simple statistical baselines
>
> ****Response 1 & 2****
>
> Thank you for your detailed and insightful feedback regarding the dataset size and inclusion of simpler baselines.
>
> For the pre-training dataset size, our primary focus was to develop a modeling strategy capable of handling wearable signals with arbitrary channel numbers and sensor types (e.g., heart, brain, physical activity signals). In response to your concern and to enhance the robustness of our approach, we have significantly expanded our pre-training and downstream datasets. This expansion addresses the generalization concern, and we have re-run all experiments to validate the effectiveness of our model. Updated results are provided in the revised manuscript, and our model consistently achieves peak performance across all scenarios. Our detailed updates are stated below:
>
> **1) Pretraining data:** Originally, for computational efficiency, we had not utilized all available samples in our pre-training dataset. To address this, we have now included all available samples and supplemented the dataset with several additional collections, including datasets containing both ECG and PPG signals, as well as a large multi-channel EEG dataset with over 50,000 samples. As a result, the pre-training dataset has grown from 37k samples (62 hours) to 241k samples (402 hours). Since there are very few publicly available multivariate sensor datasets, we further leverage time series augmentation techniques detailed in the appendix, we expanded the dataset to 2.5 million samples (approximately 4000 hours). All experiments were re-run with the model retrained on this significantly larger pre-training dataset.
>
> **2) Downstream data:** We incorporated several new datasets to further validate the model's performance. These include an EEG dataset with 500 subjects and five tasks, a dataset focused on muscular disorders, and a new ECG abnormality detection dataset. This expanded evaluation framework ensures a more comprehensive assessment of the model across diverse health-related tasks.
>
> **3) New baselines:** We have incorporated statistical baselines, CLAP, and TF-C in our evaluation to provide a more comprehensive comparison. These additional baselines offer clearer context for understanding our model's performance. Results, including comparisons with these baselines, have been updated in the manuscript as well. To further clarify our choices: modeling multivariate wearable signals, which include a wide range of modalities such as heart, brain, and physical motion signals, poses unique challenges, as no universally recognized open-source baseline or state-of-the-art model exists in this domain. To address this, we selected baselines from three representative paradigms:
>
> - **Statistical features:** Handcrafted statistical features are a traditional yet widely used method in signal processing literature. We included this baseline as a sanity check to benchmark our model against simpler approaches.
> - **SSL Time-series frameworks:** Wearable sensory data is inherently sequential, making time-series modeling strategies relevant. We compared our approach with Chronos, a state-of-the-art time-series forecasting framework, and TF-C, a commonly used self-supervised learning method for time-series data.
> - **Spectrogram-based methods:** Spectrogram-based modeling has demonstrated state-of-the-art performance in tasks such as music classification and physiological signal analysis. CLAP was selected as a representative baseline for this paradigm.
>
> **4) Linear Probing Algorithm:** To ensure better reproducibility, numerical stability, and fairness in performance comparison, we have updated our linear probe implementation. Previously, our results were based on a custom linear layer optimized with the Adam optimizer. To standardize the evaluation pipeline, we now utilize well-established methods for solving the linear probe tasks. Specifically, classification tasks are addressed using Newton's method with conjugate gradient, while regression tasks (e.g., vital signs prediction) are solved using Cholesky decomposition with a closed-form solution. These updates provide a more robust and interpretable framework for evaluating our model's representations across various tasks.
>
> Results, including comparisons with these baselines is attached in the next part of the response, and we've also updated it in the manuscript as well.

---

> ### Author Response · Authors · 2024-11-21
> **Response to Reviewer p9eE Part III**
>
> **Result table following previous part**
>
> | **Downstream Tasks**                              | **Statistical** | **Chronos** | **CLAP** | **TF-C** | **NormWear (Ours)** |
> |---------------------------------------------------|-----------------|-------------|----------|----------|---------------------|
> | WESAD                                             | 66.213          | 71.489      | 72.383   | 69.865   | **76.060**          |
> | UCI-HAR                                           | 95.784          | 91.593      | 96.420   | 96.892   | **98.954**          |
> | DriverFatigue                                     | 63.249          | **76.722**  | 61.889   | 66.882   | 74.292              |
> | **Activity Recognition Avg.**                     | 75.082          | 79.935      | 76.897   | 77.880   | **83.102**          |
> |---------------------------------------------------|-----------------|-------------|----------|----------|---------------------|
> | Epilepsy (eye open)                               | 82.489          | 82.41       | 85.094   | 89.153   | **92.743**          |
> | Epilepsy (eye close)                              | 87.457          | 88.218      | 89.867   | 94.416   | 94.828              |
> | Epilepsy (health area)                            | 86.274          | 81.08       | 83.711   | 85.619   | **88.541**          |
> | Epilepsy (tumor area)                             | 82.816          | 81.034      | 83.644   | 86.348   | 87.197              |
> | Epilepsy (seizure)                                | 88.272          | 97.572      | **97.734**| 93.998   | 97.053              |
> | GAMEEMO                                           | 51.009          | 53.747      | 52.551   | **56.275**| 54.937              |
> | **EEG Main Tasks Avg.**                           | 79.720          | 80.677      | 82.100   | 84.302   | **85.883**          |
> |---------------------------------------------------|-----------------|-------------|----------|----------|---------------------|
> | PhysioNet Challenge                               | 42.592          | 48.166      | 43.058   | **50.934**| 46.904              |
> | ECG-Abnormal                                      | 97.092          | 98.585      | 97.23    | 98.275   | **99.140**          |
> | PPG-BP (HTN)                                      | 59.499          | 52.425      | 56.757   | **65.229**| 62.341              |
> | PPG-BP (DM)                                       | 47.823          | 51.164      | 42.455   | **57.883**| 55.893              |
> | PPG-BP (CVA)                                      | **71.25**       | 50.278      | 51.667   | 58.125   | 70.625              |
> | PPG-BP (CVD)                                      | 51.219          | 58.31       | 50.91    | **58.674**| 51.773              |
> | PhysioNet EMG                                     | **99.309**      | 61.6        | 98.627   | 78.308   | 99.216              |
> | **Risk Evaluation Avg.**                          | 66.969          | 60.075      | 62.958   | 66.775   | **69.413**          |
> |---------------------------------------------------|-----------------|-------------|----------|----------|---------------------|
> | Noninvasive-BP                                    | 92.31           | 91.79       | 91.922   | 87.481   | **92.420**          |
> | PPG-Hgb                                           | 94.219          | **95.005**  | 94.291   | 93.408   | 94.632              |
> | Fetal-fPCG                                        | 98.929          | 99.048      | **99.195**| 99.077   | 99.072              |
> | **Vital Signs Avg.**                              | 95.153          | 95.281      | 95.136   | 93.322   | **95.375**          |
> |---------------------------------------------------|-----------------|-------------|----------|----------|---------------------|
> | **Micro Avg.**                                    | 76.727          | 75.276      | 76.284   | 78.255   | **80.875**          |
> | **Macro Avg.**                                    | 79.231          | 78.992      | 79.273   | 80.570   | **83.443**          |
>
> Thank you for your constructive feedback!

---

> > ### Comment · Reviewer_p9eE · 2024-11-29
> > **Response to authors**
> >
> > I appreciate the authors' detailed responses to my previous review. They've clarified the motivation behind their work and highlighted the potential contributions, particularly concerning the tokenization terminology and dataset selection.
> >
> > However, I still believe that demonstrating the generalizability of the proposed method is crucial for its acceptance. The new results still miss confidence intervals and/or statistical significance tests to assess if the model outperforms baselines. The new linear probing methods are not common in the literature, I am wondering why the authors do not simply use a linear/logistic model or a simple neural network. While I maintain my current score, I am open to reconsidering it if the authors can address these remaining concerns convincingly.

---

> > > ### Author Response · Authors · 2024-12-01
> > > **Response to Reviewer p9eE**
> > >
> > > We sincerely thank the reviewer for their continued constructive feedback and for recognizing the clarified motivation and contributions of our work.
> > >
> > > ## Clarification on the "linear probing methods":
> > > We apologize for the confusion in our initial response. To clarify, the methods we used are standard logistic regression (solved using Newton's method) and linear regression (solved via Cholesky's method).
> > > While evaluation results can naturally vary depending on the chosen methodology, the updated framework offers a more consistent basis for comparison with robustness to hyperparameters (e.g. learning rate, batch size, etc.), ensuring a fairer comparison. Notably, our model continues to perform strongly across the board, maintaining its effectiveness under this revised evaluation approach.
> > > We hope this clarification addresses the concerns about the linear prob method, and we will correct the terminology used in manuscript as well.
> > >
> > > ## Regarding statistical test:
> > > We acknowledge the importance of having statistical validation. Following the suggestion, we first run the downstream evaluation 100 times for each model on all the tasks, without fixing random seed. We observed that the outcomes stay consistent due to the stability of the optimization process.
> > >
> > > We then conduct a permutation test, across the results from these 100 runs, to assess whether our method significantly outperforms the baselines.
> > > We declare the alternative hypothesis as whether the score (AUC ROC) of NormWear is greater than the baselines in comparison. The reported P value represents the probability of observing a test statistic as extreme as, or more extreme than, the observed difference under the null hypothesis, assuming that the AUC ROC score of NormWear is not greater than the baseline.
> > > The results indicate that in nearly all cases, the statistical significance (p-value) is less than 0.01, providing statistical significance evidence of the robustness and superiority of our approach. In the following table, we include the results from conducting the statistical test across different task groups (the groups were highlighted with different colors in the tables in main sections) and the total average scores.
> > >
> > > | Ours\Baselines       | Stats | Chronos | CLAP   | TFC    |
> > > |-----------------------|-------|---------|--------|--------|
> > > |NormWear - activity   | P<.01  | P<.01    | P<.01   | P<.01   |
> > > |NormWear - eeg        | P<.01  | P<.01    | P<.01   | P<.01   |
> > > |NormWear - risk       | P<.01  | P<.01    | P<.01   | P<.01   |
> > > |NormWear - vital      | P<.01     | P<.01     | P<.01   | P<.01   |
> > > |NormWear - micro avg.     | P<.01  | P<.01    | P<.01   | P<.01   |
> > > |NormWear - macro avg.     | P<.01  | P<.01    | P<.01   | P<.01   |
> > > |-----------------------|-------|---------|--------|--------|
> > > |Conover post hoc    | <.001  | <.001    | <.001   | <.05   |
> > >
> > > We also present the critical difference diagram (CD) to visually compare the performances of multiple models across datasets, highlighting whether their performance differences are statistically significant. In order to achieve CD diagram, we first conduct Friedman Chi square test on the scores achieved by the models across all the downstream tasks, and observe P value of P < .001, making sure all the models' performance are coming from different distribution. Then we conduct Conover post hoc test to check the pair-wise model performance difference, where the P values corresponding to NormWear vs. baselines are presented in the last row of the above table. Finally, we create CD diagram based on these results, and result in the following diagram (described by text below). Our proposed model, NormWear is far apart from the bar, indicating its statistical significance against competitive baselines.
> > >
> > > ### Average score rank:
> > > | Stats       |Chronos  | CLAP | TFC   |  NormWear   |
> > > |-----------------------|-------|---------|--------|--------|
> > > |0.48   | 0.50  | 0.53    | 0.65   | **0.83**   |
> > >
> > > ### Cross bars:
> > > | 0       |1  | 2 |3 |
> > > |-----------------------|-------|---------| ---------|
> > > |(Stats, Chronos)   | (Chronos, CLAP)  | (CLAP, TFC)    | **(NormWear (ours))**|
> > >
> > >
> > > We will include the above results in the manuscript as well.
> > >
> > > We deeply appreciate your thoughtful suggestions, which have significantly contributed to enhancing this work. We remain fully open to any additional feedback that could further improve the study and better address your concerns.

---

### Author Response · Authors · 2024-11-21
**General Response**

We sincerely thank all four reviewers for their thoughtful and detailed feedback. We are delighted that our work has been recognized for its novelty, the thoughtful design of the NormWear model, its robust performance across diverse healthcare applications, and the depth of our investigations into feature visualizations, fusion techniques, and ablations. We also appreciate the reviewers’ recognition of the transparency in our communication, including our use of public datasets and an open codebase. Your positive comments are deeply encouraging and reflect the goals we aimed to achieve in this work.

In response to your feedback, we made significant revisions to the manuscript. These include expanding our pretraining data size, adding 7 new downstream tasks, and incorporating 3 additional baselines. To support these updates, we added 4 tables and 2 new figures summarizing our findings. Additionally, we made various minor revisions throughout the manuscript to address specific concerns and clarify our contributions.

We have uploaded the revised manuscript to OpenReview, with all changes highlighted in blue for transparency. The Appendix has been expanded to include further details on our experiments, and we carefully proofread the paper to fix typos and enhance clarity.

We hope our responses address all key concerns and further clarify the significance of our work. If you feel there are any remaining issues that we have not sufficiently addressed to warrant an increased score, we would greatly appreciate additional feedback on how we can further improve. Thank you again for your thoughtful and constructive comments—we deeply value your insights and hope you find our responses helpful and informative.

---

### Author Response · Authors · 2024-12-04
**Final Response**

Dear reviewers,

We sincerely thank the reviewers for their thoughtful feedback and recognition of the significance of the challenges we addressed in this work. We are particularly encouraged by the reviewers’ acknowledgment of our success in tackling key challenges in the field, mainly NormWear’s ability, **to process diverse wearable sensor signals—regardless of modality, channel count, or order**—while excelling in a **wide range of health-related tasks** highlights its adaptability and practical impact. These acknowledgments reinforce our commitment to advancing the field through impactful, well-validated research that addresses critical challenges and pushes the boundaries of healthcare applications.

At the outset, we acknowledge that our initial presentation did not fully capture NormWear's potential. Thanks to the reviewers’ insightful suggestions and guidance—including **adding stronger baselines** and **expanding the pretraining dataset size**—we are pleased to see that NormWear has withstood these rigorous evaluations, maintaining its superior performance across all tasks. We are also encouraged by the reviewers’ recognition of our extension work, as well as our contributions to the community through **open-sourcing all models and clean preprocessed data**. We are grateful for the acknowledgment that we have addressed most of the critical concerns.

In closing, we believe that the reviewers and our team share a common vision for the future of this field. We are deeply grateful for the reviewers’ guidance, which has been invaluable to us. Through this process, we have envisioned the potential of a unified model for physiological sensors and hope that our efforts will ultimately contribute to the growth and advancement of this promising direction.

Best,
Authors

---

### Meta-Review · Area_Chair_NtWQ · 2024-12-08

**Metareview:**

(a) Summary of Scientific Claims and Findings:

The paper introduces NORMWEAR, a foundation model tailored for multi-modal, wearable-sensing healthcare data. It employs a novel tokenization strategy that efficiently encodes heterogeneous biosignals and a channel-aware attention mechanism to handle diverse input modalities, including ECG, EEG, and PPG. By aligning representations for zero-shot inference, NORMWEAR can apply learned knowledge to new tasks without retraining. The authors demonstrate the model’s versatility and improved performance over standard baselines through evaluations on 12 distinct downstream health-related tasks.

(b) Strengths of the Paper:

The proposed fusion and tokenization approaches are well-justified, indicating careful consideration of the complexities inherent in multi-modal biosignals.
The model shows robust performance across a wide range of heterogeneous tasks, suggesting strong generalizability and zero-shot capabilities.
The extensive ablations highlight the contributions of individual components, underscoring the thoughtful design choices and their empirical support.

(c) Weaknesses and Missing Elements:

The semantic alignment training procedure for zero-shot inference is not clearly explained, leaving questions about how embeddings are aligned with semantic text and what datasets are used for this step.
Baselines may not be optimally chosen or described for certain tasks, reducing the clarity of the model’s relative performance.
Key aspects of the Memory Stream Inspired Mechanism (MSiTF), including the motivation for human-centric optimization and the interpretation of various scoring functions (e.g., recency, importancy scores), remain insufficiently detailed.
The rationale behind using both L1 loss and cosine distance is not thoroughly justified, and the provided visualizations need more interpretive guidance.
The paper lacks a dedicated prior work section, making it difficult to situate NORMWEAR within the broader foundation model literature.

(d) Reasons for Decision (Accept/Reject):

On the positive side, NORMWEAR addresses a significant need—handling complex, heterogeneous biosignals across a variety of healthcare tasks—and provides promising evidence of strong performance and generalizability. The methodological innovations, particularly the tokenization and fusion mechanisms, are valuable contributions that could inspire future work in multimodal healthcare analytics. However, the paper leaves important aspects underexplained, notably the semantic alignment procedure and certain architectural components. The absence of a dedicated prior work comparison also hinders a full appreciation of the novelty and positioning of NORMWEAR relative to the state of the art. I think it has some merits but hasn't achieved the acceptance bar.

**Additional Comments On Reviewer Discussion:**

Main issues:
- **Semantic alignment training details unclear:** Uncertainty about how embeddings are aligned with semantic text and what datasets are used. References provided do not clarify a specific approach.
- **Insufficient baselines:** The chosen baselines (e.g., Chronos) are not directly suited for the downstream tasks, and the ViT baseline’s zero-shot setup lacks clarity.
- **Unclear MSiTF mechanism:** The human-sensing optimization, the use of standard deviation/mean, recency score interpretation, and the gating mechanism (importancy score) are not well explained.
- **Loss function rationale unclear:** The choice of using both L1 loss and cosine distance lacks justification.
- **Limited experimental details:** No comparison of the proposed tokenizer against simpler methods, only one metric is reported, and visualizations (t-SNE clusters, intermediate layer outputs) lack deeper interpretation.
- **No prior work section:** Difficult to understand how this approach fits into the broader foundation model landscape.

- In response to the feedback, the authors have:
  - Expanded the pretraining data size.
  - Added 7 new downstream tasks and 3 new baseline methods.
  - Included 4 new tables and 2 new figures to present results clearly.
  - Made various minor revisions to improve clarity and address specific concerns.

---

### Decision · Program_Chairs · 2025-01-22

Reject